# SS-TPT: Stability and Suitability-Guided Test-Time Prompt Tuning for Adversarially Robust Vision-Language Models

**Sunoh Kim**[1]   **Daeho Um**[2]

## Abstract

Vision-language models (VLMs) such as CLIP achieve strong zero-shot recognition but remain highly fragile under adversarial perturbations. Recent test-time adaptation defenses improve robustness by leveraging many augmented views, but this leads to impractical slowdown and a clear robustness-throughput trade-off. To address this challenge, we present Stability and Suitability-guided Test-time Prompt Tuning (SS-TPT), evaluating the quality of each augmented view via two complementary scores: (1) stability, measuring prediction invariance to weak augmentations, and (2) suitability, measuring feature-space density among views. These stability and suitability (SS) scores guide both adaptation and inference through an SS-guided consistency loss and an SS-weighted prediction, amplifying trustworthy views while suppressing corrupted ones. Extensive experiments demonstrate that SS-TPT significantly outperforms prior state-of-the-art methods, achieving superior robustness-throughput trade-offs across diverse datasets and varying numbers of views, thereby demonstrating both strong practicality and generality. Our code is available at https://github.com/sunoh-kim/SS-TPT.

## 1. Introduction

Vision-language models (VLMs) (Radford et al., 2021; Jia et al., 2021; Yu et al., 2022) are pretrained on large-scale image-text pairs, enabling strong zero-shot inference across diverse downstream tasks. By aligning visual and textual modalities through contrastive training, CLIP (Radford et al., 2021) has become a milestone VLM with a concise architec-

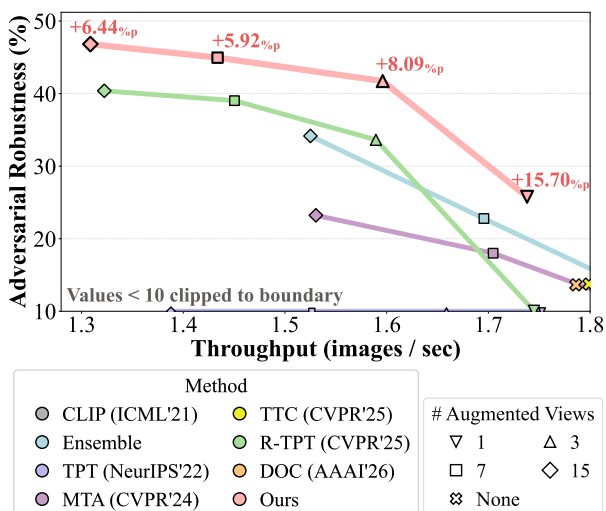

*Figure 1.* Robustness-throughput trade-off across test-time defenses, averaged over 10 fine-grained classification datasets. Unlike prior methods that either sacrifice computational efficiency or robustness, our approach consistently achieves the best of both worlds, delivering higher robustness at faster speeds, even with few augmented views. $+\Delta\%p$ indicates the robustness improvement over the strongest previous method at the identical view setting.

ture and strong performance, and has been widely adopted in both research and applications (Wang et al., 2024b; Zhong et al., 2022; Yao et al., 2023). However, CLIP and its variants remain vulnerable to distribution shifts, *i.e.*, mismatches between the training and testing data distributions. To deal with distribution shifts, test-time adaptation (TTA) has been proposed, which updates model behavior at inference without labeled data (Zhang et al., 2022; 2024d; Farina et al., 2024). TTA aligns predictions with the target distribution and requires no task-specific retraining. Nevertheless, many studies (Li et al., 2024; Mao et al., 2023; Zhang et al., 2024c; Zhou et al., 2024) show that CLIP remains susceptible to adversarial attacks (Szegedy et al., 2014; Goodfellow et al., 2015), motivating defensive TTA methods that explicitly target adversarial robustness. As vision-language models become increasingly integrated into real-world applications, developing such defensive TTA methods has become essential to ensure trustworthy deployment.

Recent defensive TTA methods for VLMs (Jiang et al., 2026;

[1]Dankook University, Yongin, South Korea [2]University of Seoul, Seoul, South Korea. Correspondence to: Daeho Um <daehoum@uos.ac.kr>.

*Proceedings of the 43rd International Conference on Machine Learning*, Seoul, South Korea. PMLR 306, 2026. Copyright 2026 by the author(s).

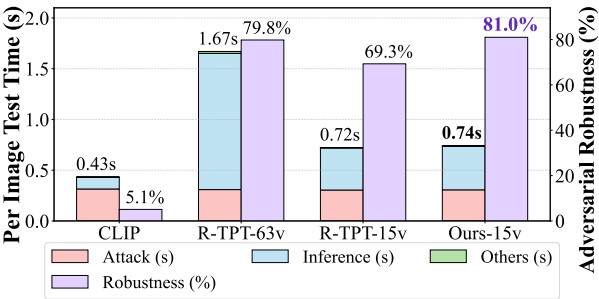

*Figure 2.* Comparison of per-image test time and adversarial robustness on the Caltech101 dataset across CLIP, R-TPT with different numbers of augmented views (63 and 15 views), and our method with 15 views. These results highlight the trade-off that more views improve robustness but slow down inference, while our approach achieves strong robustness with fast inference.

Wang et al., 2025; Sheng et al., 2025) commonly rely on many augmented views, which are diverse transformations of the original image, to stabilize adaptation. Among them, inspired by test-time prompt tuning (TPT) (Shu et al., 2022), R-TPT (Sheng et al., 2025) achieves stronger robustness by tuning textual prompts for adaptation and aggregating predictions from augmented views for inference. However, the computational cost of these defensive TTA methods grows with the number of views, creating a substantial throughput gap from vanilla CLIP, the underlying backbone of these defenses. For instance, as shown in Fig. 2, CLIP processes an image in 0.43s, whereas R-TPT-63v requires 1.67s, which is nearly *4 times slower* than CLIP, severely undermining its practicality. A natural idea is therefore to reduce the number of views. However, since most existing defenses assume that all views are equally trustworthy during the adaptation process, using fewer views amplifies the influence of any corrupted view, which in turn degrades robustness. As shown in Fig. 2, reducing R-TPT from 63 to 15 views speeds up inference but *lowers robustness by over 10%*, revealing a clear efficiency-robustness trade-off.

In this paper, we empirically identify that this vulnerability arises from the lack of mechanisms to assess the *quality* of the views, since a single low-quality view can dominate adaptation in the few-view regime. Building on this observation, we propose **Stability- and Suitability-guided Test-time Prompt Tuning (SS-TPT)**, which leverages meaningful scores that measure view quality and guide both adaptation and inference. These complementary scores are *stability* and *suitability*, computed from multi-view information. Specifically, stability is defined as the invariance of a view's predictive distribution when the view is perturbed by weak stochastic augmentations, and suitability is defined as the density of a view's surrounding views in the feature space. These two scores serve as key indicators of trustworthy views, distinguishing them from corrupted or outlier views. Using these scores, SS-TPT (i) applies an *SS-guided*

*consistency loss* during adaptation to align predictions with trustworthy references, and (ii) performs an *SS-weighted prediction* during inference to aggregate per-view predictions into the final output that emphasizes trustworthy views while suppressing corrupted ones.

By focusing on view quality over quantity, SS-TPT outperforms prior methods in both robustness and efficiency. As shown in Fig. 1, SS-TPT consistently delivers superior robustness-throughput trade-offs across a wide range of augmented view counts, highlighting its practicality. Remarkably, even under the extremely low visual diversity regime (one augmented view), SS-TPT maintains high robustness, underscoring its critical role in extracting meaningful information from limited visual evidence.

In summary, our contributions are as follows:

- We introduce **SS-TPT**, a test-time prompt tuning defense for CLIP that computes per-view *stability* and *suitability* (SS) scores to measure view quality and guide both adaptation and inference.

- We design adaptation and inference mechanisms leveraging SS scores: an *SS-guided consistency loss* that enforces trustworthy alignment, and an *SS-weighted prediction* that suppresses outliers for final prediction.

- We demonstrate that SS-TPT achieves state-of-the-art performance in test-time defenses and superior robustness-throughput trade-offs across various settings, including diverse datasets, distribution shifts, attack scenarios, and varying numbers of views, highlighting SS-TPT's strong practicality and generality.

## 2. Related Work

### 2.1. Adversarial Attacks and Defenses

Small and often imperceptible perturbations to input images can cause models to make incorrect predictions, known as adversarial attacks (Szegedy et al., 2014; Goodfellow et al., 2015). In white-box settings, projected gradient descent (PGD) serves as the standard benchmark (Madry et al., 2018), while the Carlini-Wagner (CW) attack (Carlini & Wagner, 2017) and AutoAttack (Croce & Hein, 2020) remain strong baselines. Beyond these, diverse attack paradigms have been explored, including universal (Moosavi-Dezfooli et al., 2017), black-box (Ilyas et al., 2018; Andriushchenko et al., 2020), and transferability-oriented attacks (Xie et al., 2019; 2025). On the defense side, adversarial training remains dominant (Madry et al., 2018; Zhang et al., 2019; Wu et al., 2020), while purification via generative models is also explored (Samangouei et al., 2018; Nie et al., 2022). While most defenses focus on

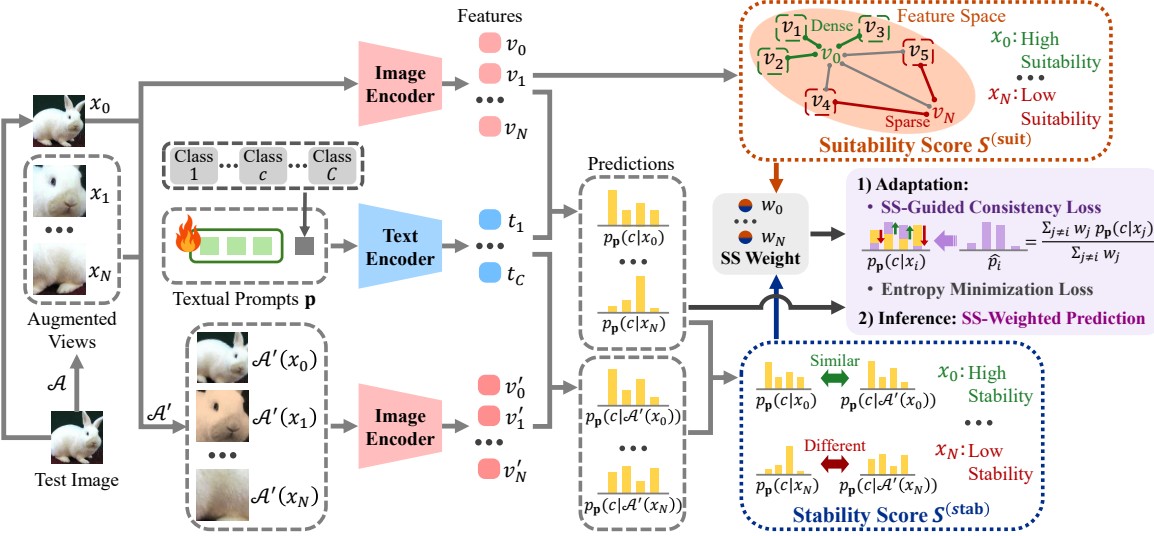

*Figure 3.* Overview of SS-TPT. From a test image, multiple views are generated through augmentations $\mathcal{A}$. Each view is evaluated by two quality scores: (i) *stability*, measuring invariance under weak augmentations $\mathcal{A}'$, and (ii) *suitability*, assessing feature-space density among views. These scores produce SS weights that guide both adaptation and inference: (1) an SS-guided consistency loss aligns predictions toward more trustworthy views, and (2) an SS-weighted prediction aggregates predictions while suppressing corrupted views.

vision-only models, recent work extends adversarial robustness to vision-language models (VLMs). This direction is particularly important because VLMs serve as foundation models for a broad range of vision-language applications, including detection and grounding (Zhong et al., 2022; Yao et al., 2023; Kim et al., 2022; 2024a;b; 2026). Training-based defenses include contrastive adversarial training on CLIP (Mao et al., 2023; Schlarmann et al., 2024) as well as adversarial prompt tuning (Li et al., 2024), though these require labeled data or task-specific tuning. In contrast, recent test-time defenses (Wang et al., 2025; Sheng et al., 2025) adapt prompts without any task-specific supervision. Building on this paradigm, we introduce a test-time defense for VLMs that explicitly evaluates view trustworthiness, enabling robust adaptation across various benchmarks.

## 2.2. Test-Time Adaptation and Prompt Tuning

Test-time adaptation (TTA) mitigates distribution shift by updating the model at inference. Existing methods range from semantic to efficiency-oriented strategies (Wang et al., 2021; Farina et al., 2024; Zhang et al., 2024d; 2022; Hübotter et al., 2025). Recent work has investigated adversarial robustness in zero-shot CLIP (Xing et al., 2025; Tong et al., 2025; Zanella & Ben Ayed, 2024; Jiang et al., 2026; Kim & Um, 2026), using counterattacks or augmentations. As an alternative to full fine-tuning, prompt tuning has emerged in VLMs (Zhou et al., 2022; Li et al., 2024; Zhang et al., 2024c; Zhou et al., 2024). Building on CLIP, Shu *et al.* (Shu et al., 2022) introduced Test-time Prompt Tuning (TPT), which adapts textual prompts for each instance. Subsequent work has expanded TPT with diverse augmentations (Ma

et al., 2023), calibration (Yoon et al., 2024; Sharifdeen et al., 2025), historical memory (Zhang et al., 2024b), reward-based objectives (Zhao et al., 2024), and robustness-oriented designs (Zhang et al., 2024a; Wang et al., 2024a; Jeong et al., 2025). Other studies explore dynamic updates (Xiao et al., 2025) as well as black-box (Meng et al., 2025), or hierarchical (Zhang et al., 2025; Wu et al., 2025). Collectively, these efforts establish TPT as a versatile paradigm in VLMs. For adversarial robustness, R-TPT (Sheng et al., 2025) performs point-wise entropy minimization with a reliability-based ensemble. In contrast, our SS-TPT explicitly quantifies view quality and leverages it to guide both adaptation and inference, achieving strong robustness even with few views.

## 3. Methodology

### 3.1. Overview of SS-TPT

We propose Stability and Suitability-guided Test-time Prompt Tuning (SS-TPT), a test-time adaptation framework for CLIP. Fig. 3 provides an overview of SS-TPT. To measure the quality of each augmented view, SS-TPT uses two complementary scores from views: stability and suitability. Stability captures how invariant a view's predictions remain under weak stochastic augmentations, while suitability reflects how densely a view is surrounded by other views in the feature space. These scores jointly guide adaptation and inference via an SS-guided consistency loss and an SS-weighted prediction, respectively. For adaptation, the SS-guided consistency loss leverages the consistency term that enforces alignment with trustworthy views. For robust inference, the SS-weighted prediction aggregates predic-

tions while downweighting noisy or adversarial views.

## 3.2. Preliminaries

**CLIP classifier.** CLIP (Radford et al., 2021) is a vision-language model that aligns images and texts in a joint embedding space. In a classification task with label set $\{y_c\}_{c=1}^C$, the text encoder $G(\cdot)$ maps the class name $y_c$ into a textual feature $\mathbf{t}_c = G(\mathrm{prompt}(y_c))$, where $\mathrm{prompt}(\cdot)$ denotes a fixed template. The image encoder $F(\cdot)$ maps an input image $x$ into a visual feature $\mathbf{v} = F(x)$. The likelihood of assigning $x$ to class $c$ is computed as

$$p(c \mid x) = \frac{\exp\big(\cos(\mathbf{v}, \mathbf{t}_c)/\tau\big)}{\sum_{j=1}^C \exp\big(\cos(\mathbf{v}, \mathbf{t}_j)/\tau\big)}, \quad (1)$$

where $\cos(\cdot, \cdot)$ denotes the cosine similarity and $\tau$ is a temperature constant.

**Test-time prompt tuning objective.** While CLIP achieves strong zero-shot recognition, its accuracy can degrade under distribution shifts. To address this limitation, test-time prompt tuning (TPT) (Shu et al., 2022) was proposed, which adapts prompts during test time. Given a test image $x_0$, TPT generates a set of views through augmentations $\mathcal{A}$, including the original $x_0$ and its augmented views, and selects a low-entropy subset $\mathcal{B}$. The adaptation objective minimizes the marginal entropy of averaged predictions:

$$\mathcal{L}_{\mathrm{marginal}} = \mathcal{H}\big(\frac{1}{|\mathcal{B}|} \sum_{x_i \in \mathcal{B}} p_{\mathbf{p}}(\cdot \mid x_i)\big), \quad (2)$$

where $\mathcal{H}(\cdot)$ denotes the Shannon entropy, $p_{\mathbf{p}}(\cdot \mid x_i)$ represents the predicted probability distribution over all classes for $x_i$, and $\mathbf{p}$ refers to the learnable prompt parameters that condition the text encoder. This can be decomposed into $\mathcal{L}_{\mathrm{marginal}} = \frac{1}{|\mathcal{B}|} \sum_{x_i \in \mathcal{B}} \mathcal{H}\big(p_{\mathbf{p}}(\cdot|x_i)\big) + \frac{1}{|\mathcal{B}|} \sum_{x_i \in \mathcal{B}} D_{\mathrm{KL}}\big(p_{\mathbf{p}}(\cdot|x_i) \,\|\, \bar{p}\big)$, where $D_{\mathrm{KL}}(\cdot\|\cdot)$ denotes the Kullback-Leibler (KL) divergence and $\bar{p}$ is the mean prediction over $\mathcal{B}$. To improve robustness under adversarial attacks, R-TPT (Sheng et al., 2025) removes the KL term, since enforcing consistency with corrupted views can mislead adaptation, and retains only the point-wise entropy:

$$\mathcal{L}_{\mathrm{point}} = \frac{1}{|\mathcal{B}|} \sum_{x_i \in \mathcal{B}} \mathcal{H}\big(p_{\mathbf{p}}(\cdot \mid x_i)\big). \quad (3)$$

## 3.3. Stability and Suitability

In contrast to previous TPT techniques (Shu et al., 2022; Sheng et al., 2025), we introduce two scores to quantify the quality of each view $x_i$: *stability* $S_i^{\mathrm{stab}}$ and *suitability* $S_i^{\mathrm{suit}}$. These scores jointly guide the adaptation by enabling consistency with trustworthy views through a KL term, and improve prediction robustness by weighting views.

**Stability under weak augmentations.** For each view, we expect the prediction to remain consistent under weak augmentations. Let $\mathcal{A}'$ denote a weak augmentation distribution comprising affine transformations, color jitter, blur, and noise, each applied stochastically. Given the base prediction on a view $x_i$, *i.e.*, the probability distribution $p_{\mathbf{p}}(\cdot \mid x_i)$, we define its stability as the inverse of the divergence to predictions of its augmented variant $\tilde{x}_i \sim \mathcal{A}'(x_i)$:

$$S_i^{\mathrm{stab}} = \Big(D_{\mathrm{JS}}\big(p_{\mathbf{p}}(\cdot \mid x_i) \,\|\, p_{\mathbf{p}}(\cdot \mid \tilde{x}_i)\big) + \eta\Big)^{-1}, \quad (4)$$

where $D_{\mathrm{JS}}(p\|q) = \frac{1}{2} D_{\mathrm{KL}}\big(p \,\|\, \frac{p+q}{2}\big) + \frac{1}{2} D_{\mathrm{KL}}\big(q \,\|\, \frac{p+q}{2}\big)$ denotes the Jensen-Shannon divergence, and $\eta$ is a small constant added to avoid division by zero. A larger value of $S_i^{\mathrm{stab}}$ indicates greater prediction invariance under weak augmentations, *i.e.*, higher stability. Intuitively, a low stability indicates that the prediction fluctuates considerably across an augmented variant of the same image, suggesting unreliable behavior under minor perturbations.

**Suitability in the feature space among views.** For each view, we expect its representation to lie in a dense region formed by the other views in the feature space, ensuring that outliers are suppressed. Let $\mathbf{v}_i = F(x_i)/\|F(x_i)\|$ be the normalized feature and $s_{ij} = \mathbf{v}_i^\top \mathbf{v}_j$ the cosine similarity. We define a cosine-induced distance $d_{ij} = 2 - 2s_{ij}$, which is derived as $d_{ij} = \|\mathbf{v}_i - \mathbf{v}_j\|^2 = \|\mathbf{v}_i\|^2 + \|\mathbf{v}_j\|^2 - 2\mathbf{v}_i^\top \mathbf{v}_j = 2 - 2s_{ij}$. Intuitively, if $x_i$ lies in a dense feature space, its distances to the other views $d_{ij}$ are relatively small. Such densely clustered views are generally more suitable as valid views. We quantify this suitability by inverse mean distance:

$$S_i^{\mathrm{suit}} = \Big(\frac{1}{|\mathcal{B}| - 1} \sum_{j \in \mathcal{B}, \, j \neq i} d_{ij} + \eta\Big)^{-1}, \quad (5)$$

where $\eta$ is a small constant added to avoid division by zero. A larger value of $S_i^{\mathrm{suit}}$ indicates that $x_i$ lies in a high-density region, *i.e.*, higher suitability. Intuitively, a low suitability indicates that the view is inconsistent with other views in the feature space, suggesting an outlier or a corrupted view.

**Stability-Suitability (SS) weighting.** To represent the overall quality of each view, we combine the stability and suitability scores into a *Stability-Suitability (SS)* score, balancing each view's invariance under augmentations and feature-space density among views. After min-max normalization, *i.e.*, $\bar{S}_i^{\mathrm{suit}}, \bar{S}_i^{\mathrm{stab}} \in [0, 1]$, the SS score $Z_i$ is computed as $Z_i = \alpha \bar{S}_i^{\mathrm{suit}} + (1 - \alpha) \bar{S}_i^{\mathrm{stab}} \in [0, 1]$, where $\alpha \in [0, 1]$ is a hyperparameter controlling the trade-off between stability and suitability. $Z_i$ is then converted into Stability-Suitability (SS) weights via a softmax:

$$w_i = \frac{\exp(Z_i/\tau_w)}{\sum_j \exp(Z_j/\tau_w)}, \quad (6)$$

where $\tau_w$ is a temperature constant, set to 0.25.

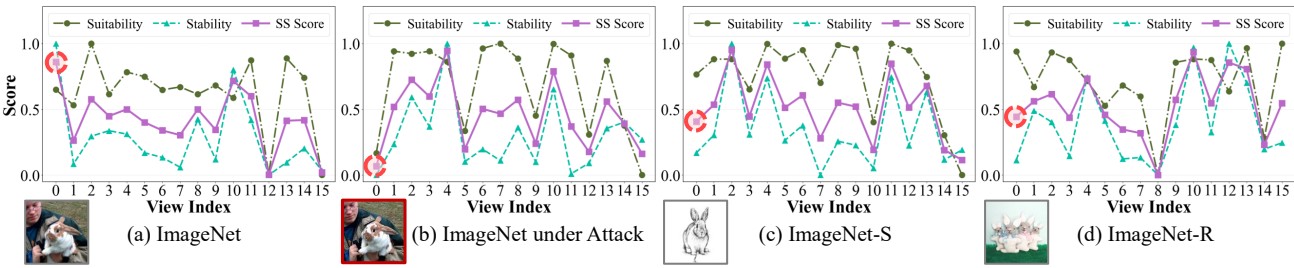

**Figure 4.** Stability scores, suitability scores, and combined SS scores for the original view (index 0, indicated by the red dot) and its augmented views on (a) clean ImageNet, (b) ImageNet under attack, and (c-d) distribution-shifted variants: ImageNet-S and ImageNet-R. The natural original in (a) has the highest SS score, while the perturbed original in (b) has a near-zero score, and the distribution-shifted originals in (c-d) receive lower scores compared to the natural original in (a).

**Impact of SS weighting.** Fig. 4 presents stability, suitability, and combined SS scores across different conditions. In the clean ImageNet, the SS score is highest at the original view (the red dot in Fig. 4a), indicating that SS scoring focuses on the natural view. In contrast, under adversarial attack or distribution shift, SS scoring downweights the original (the red dots in Fig. 4b-d) and instead emphasizes other augmented views. These results demonstrate that SS scoring prioritizes trustworthy views across diverse conditions.

### 3.4. SS-Guided Consistency Loss

For adaptation, we introduce a test-time consistency loss that uses stable and suitable views as references, thereby enforcing alignment with trustworthy predictions while avoiding the propagation of corrupted ones. Given the SS-weights $w_i$ from Eq. (6), each view $x_i$ is aligned to a leave-one-out weighted reference, avoiding trivial self-alignment:

$$\widehat{p}^{(-i)} = \frac{\sum\limits_{j \neq i} w_j \, p_{\mathbf{p}}(\cdot \mid x_j)}{\sum\limits_{j \neq i} w_j}. \qquad (7)$$

The SS-guided consistency loss is then defined as

$$\mathcal{L}_{\text{scons}} = \frac{1}{|\mathcal{B}|} \sum_{x_i \in \mathcal{B}} D_{\text{KL}}\big(p_{\mathbf{p}}(\cdot \mid x_i) \,\|\, \widehat{p}^{(-i)}\big). \qquad (8)$$

**Adaptation objective.** We combine point-wise entropy minimization with the SS-guided consistency: $\mathcal{L}_{\text{adapt}} = \frac{1}{|\mathcal{B}|} \sum_{x \in \mathcal{B}} \mathcal{H}(p_{\mathbf{p}}(\cdot \mid x)) + \lambda \mathcal{L}_{\text{scons}}$, where $\lambda \geq 0$ balances the two terms. The tuned prompts, denoted as $\mathbf{p}^{\star}$, are obtained by minimizing the adaptation objective via gradient descent: $\mathbf{p} \leftarrow \mathbf{p} - \gamma \nabla_{\mathbf{p}} \mathcal{L}_{\text{adapt}}$ with the learning rate $\gamma$.

### 3.5. Analysis of SS-Guided Consistency

SS-TPT provides a new perspective on the role of consistency in test-time prompt tuning under adversarial attacks. Existing approaches can be interpreted through three paradigms.

---

**Algorithm 1** Pseudocode of SS-TPT

**Input:** Test image $x_0$, CLIP $(F, G)$, learning rate $\gamma$, loss weight $\lambda$, SS trade-off $\alpha$, temperature $\tau_w$
**Output:** Predicted class probability $\widetilde{p}(\cdot \mid x_0)$

1: Generate views $\{x_i\}$ and select subset $\mathcal{B}$.
2: Initialize prompt $\mathbf{p}$. Compute normalized features $\mathbf{v}_i = \frac{F(x_i)}{\|F(x_i)\|}, \forall x_i$.
3: **for** each view $x_i$ **do**
4:     Sample variant $\tilde{x}_i$. Compute **Stability**:
5:     $S_i^{\text{stab}} = \big(D_{\text{JS}}(p_{\mathbf{p}}(\cdot \mid x_i) \,\|\, p_{\mathbf{p}}(\cdot \mid \tilde{x}_i)) + \eta\big)^{-1}$
6:     Compute **Suitability**:
7:     $S_i^{\text{suit}} = \big(\frac{1}{|\mathcal{B}|-1} \sum_{j \neq i} (2 - 2\mathbf{v}_i^{\top} \mathbf{v}_j) + \eta\big)^{-1}$
8: **end for**
9: Normalize scores to obtain $\bar{S}_i^{\text{stab}}, \bar{S}_i^{\text{suit}} \in [0, 1]$.
10: Compute weights $w_i = \frac{\exp(Z_i/\tau_w)}{\sum_j \exp(Z_j/\tau_w)}$, where $Z_i = \alpha \bar{S}_i^{\text{suit}} + (1-\alpha)\bar{S}_i^{\text{stab}}$.
11: Compute reference $\widehat{p}^{(-i)} = \frac{\sum_{j \neq i} w_j p_{\mathbf{p}}(\cdot \mid x_j)}{\sum_{j \neq i} w_j} \forall x_i$.
12: Update prompt: $\mathbf{p} \leftarrow \mathbf{p} - \gamma \nabla_{\mathbf{p}} \mathcal{L}_{\text{adapt}}$, where $\mathcal{L}_{\text{adapt}} = \frac{1}{|\mathcal{B}|} \sum_{x_i \in \mathcal{B}} \big[\mathcal{H}(p_{\mathbf{p}}(\cdot \mid x_i)) + \lambda D_{\text{KL}}(p_{\mathbf{p}}(\cdot \mid x_i) \| \widehat{p}^{(-i)})\big]$
13: **Return** $\widetilde{p}(\cdot \mid x_0) = \sum_{x_i} w_i p_{\mathbf{p}^{\star}}(\cdot \mid x_i)$.

---

**Unguided consistency.** The foundational method, TPT (Shu et al., 2022), minimizes marginal entropy in Eq. (2), which implicitly includes an unguided KL consistency term that enforces alignment across all views, regardless of their individual quality.

**No consistency.** R-TPT (Sheng et al., 2025) argues that enforcing consistency with corrupted adversarial views can actively mislead adaptation. Consequently, R-TPT completely removes the KL term and relies exclusively on point-wise entropy, as shown in Eq. (3).

**SS-guided consistency.** Our originality stems from the observation that consistency itself is not the problem. We challenge the no-consistency paradigm by reinstating the KL term in a guided manner. Using the SS weights in Eq. (6), we formulate an SS-guided consistency loss in Eq. (8) and combine it with point-wise entropy for adaptation. This ensures that the model aligns only with trustworthy references,

*Table 1.* Comparison of clean accuracy (Acc.) and adversarial accuracy (Rob., $\epsilon = 1.0$) of various adaptation methods on 10 fine-grained datasets. Bold indicates the best performance, and underline indicates the second-best.

| Method | SUN397 | | Food101 | | Caltech101 | | DTD | | Flower102 | | Pets | | UCF101 | | Aircraft | | EuroSAT | | Cars | | Average | |
|---|---|---|---|---|---|---|---|---|---|---|---|---|---|---|---|---|---|---|---|---|---|---|
| | Acc. | Rob. | Acc. | Rob. | Acc. | Rob. | Acc. | Rob. | Acc. | Rob. | Acc. | Rob. | Acc. | Rob. | Acc. | Rob. | Acc. | Rob. | Acc. | Rob. | Acc. | Rob. |
| CLIP | 58.8 | 0.1 | 73.9 | 0.0 | 85.7 | 5.2 | 40.4 | 2.1 | 61.7 | 0.0 | 83.6 | 0.1 | 58.9 | 0.1 | 15.7 | 0.0 | 23.7 | 0.0 | 55.8 | 0.0 | 55.8 | 0.8 |
| TPT | 60.0 | 0.4 | 74.2 | 0.1 | 87.1 | 6.8 | 41.3 | 4.5 | 61.4 | 0.0 | 83.6 | 0.1 | 59.5 | 0.3 | 16.9 | 0.0 | 28.5 | 0.1 | 57.5 | 0.0 | 57.0 | 1.2 |
| Ensemble | 58.3 | 37.9 | 68.5 | 40.2 | 83.6 | 67.9 | 37.8 | 25.8 | 58.1 | 37.4 | 82.4 | 59.7 | 54.0 | 35.0 | 16.4 | 6.3 | 17.1 | 8.5 | 56.1 | 22.9 | 53.2 | 34.2 |
| TTC | 55.1 | 8.9 | 70.5 | 7.2 | 82.0 | 47.4 | 38.6 | 15.2 | 55.8 | 7.5 | 76.0 | 10.7 | 52.7 | 21.6 | 12.1 | 0.7 | 23.9 | 14.2 | 52.6 | 4.0 | 51.9 | 13.7 |
| DOC | 55.8 | 9.0 | 70.7 | 7.5 | 81.7 | 46.7 | 38.5 | 15.5 | 56.1 | 7.2 | 75.3 | 10.3 | 53.2 | 21.0 | 12.5 | 0.8 | 24.4 | 14.3 | 52.9 | 4.0 | 52.1 | 13.6 |
| R-TPT | 58.9 | 43.3 | 69.8 | 50.1 | 83.5 | 69.3 | 39.1 | 27.1 | 58.8 | 43.5 | 82.6 | 68.8 | 57.1 | 40.5 | 15.7 | 11.0 | 22.6 | 15.9 | 53.5 | 34.5 | 54.2 | 40.4 |
| Ours | 59.9 | 51.2 | 70.1 | 56.2 | 85.6 | 81.0 | 40.0 | 34.6 | 58.8 | 50.6 | 83.5 | 72.8 | 58.1 | 49.1 | 16.6 | 13.3 | 22.4 | 19.0 | 56.0 | 40.2 | 55.1 | 46.8 |

avoiding corrupted views.

### 3.6. SS-Weighted Prediction

After adaptation, we obtain tuned prompts $\mathbf{p}^\star$ and recompute per-view predictions $p_{\mathbf{p}^\star}(\cdot|x_i)$. We aggregate predictions with SS-weights from Eq. (6) over all views:

$$\widetilde{p}(\cdot \,|\, x_0) = \sum_{x_i} w_i \, p_{\mathbf{p}^\star}(\cdot \,|\, x_i). \qquad (9)$$

The SS-weighted prediction downweights corrupted views while emphasizing stable and suitable ones, yielding more robust predictions than a simple average.

## 4. Experiment

### 4.1. Experimental Setup

**Datasets.** We evaluate our method across a diverse set of benchmarks that span general objects, fine-grained objects, textures, scenes, and actions. Specifically, we adopt the following datasets: SUN397 (Xiao et al., 2010), Food101 (Bossard et al., 2014), Caltech101 (Fei-Fei et al., 2004), DTD (Cimpoi et al., 2014), Flower102 (Nilsback & Zisserman, 2008), Pets (Parkhi et al., 2012), UCF101 (Soomro et al., 2012), Aircraft (Maji et al., 2013), EuroSAT (Helber et al., 2019), and Cars (Krause et al., 2013). For large-scale evaluation, we further include ImageNet (Deng et al., 2009) and four ImageNet distribution-shift (OOD) benchmarks: ImageNet-A (Hendrycks et al., 2021b), ImageNet-V2 (Recht et al., 2019), ImageNet-R (Hendrycks et al., 2021a), and ImageNet-S (Wang et al., 2019). As we perform test-time prompt tuning, only test sets are used without any labeled data.

**Evaluation Metrics.** Following evaluation protocol of (Shu et al., 2022; Sheng et al., 2025), we report two evaluation metrics: (1) average accuracy on clean samples (Acc.) and (2) average accuracy on adversarial samples (Rob.).

**Baselines.** We compare our method against representative test-time adaptation baselines for CLIP. Specifically, we include the zero-shot prediction of CLIP (Radford et al., 2021) and TPT (Shu et al., 2022). We also adopt Ensemble, which simply averages the predictions across multiple augmented views. For adaptation-based defenses, we con-

sider TTC (Xing et al., 2025), DOC (Jiang et al., 2026), and R-TPT (Sheng et al., 2025), which represent the state-of-the-art in adversarial robustness for CLIP. All baseline results are reproduced using the official implementations. All methods are evaluated under a test-time instance-level adaptation regime, in which each test sample is adapted and predicted independently. For methods that do not use augmented views, such as TTC and DOC, we set the effective batch size to one.

**Implementation Details.** Since our tasks focus on instance-level test-time adaptation, the model update and prediction of each test sample are performed independently, without leveraging information from other samples. We adopt pretrained CLIP-ResNet50 and CLIP-ViT-B/16 (Radford et al., 2021) as the backbones. To study the effect of visual diversity, we vary the number of augmented views $N \in \{1, 3, 7, 15\}$ for both our method and all baselines, which corresponds to an effective batch size of $\{2, 4, 8, 16\}$ when including the original view. Given the generated views, we construct the subset $\mathcal{B}$ by selecting the three views with the lowest prediction entropy, or all available views when fewer than three views are available. While the subset $\mathcal{B}$ is used only for test-time adaptation, the final prediction is obtained by applying SS-weighted aggregation over all generated $N + 1$ views, following the R-TPT protocol (Sheng et al., 2025). Unless otherwise specified, all experiments are conducted using CLIP-ResNet50 with 15 augmented views as the default setting.

We follow the TPT standard (Shu et al., 2022) for adaptation settings. We set the text prompt template as "a photo of a". During adaptation, only the text prompt parameters are updated, where 4 learnable context tokens are optimized by Adam (Kingma & Ba, 2014) for a single adaptation step with a learning rate of $5 \times 10^{-3}$. For adversarial evaluation, to ensure a fair and rigorous comparison, we follow the attack protocols established in R-TPT (Sheng et al., 2025). Adversarial perturbations are generated by PGD (Madry et al., 2018): $\epsilon = 1.0$ with 7 steps for CLIP-ResNet50, and $\epsilon = 4.0$ with 100 steps for CLIP-ViT. Suitability scores are combined with stability scores via a trade-off parameter $\alpha = 0.4$. The loss balance $\lambda$ is set to 1. Notably, SS-TPT uses the same hyperparameters across fine-grained classification datasets, distribution-shift benchmarks, attack sce-

*Table 2.* Comparison of clean accuracy (Acc.) and adversarial accuracy (Rob., $\epsilon = 1.0$) of various adaptation methods on ImageNet and its four distribution-shifted variants.

| Method | ImageNet | | ImageNet-A | | ImageNet-V2 | | ImageNet-R | | ImageNet-S | | OOD Average | | Average | |
|---|---|---|---|---|---|---|---|---|---|---|---|---|---|---|
| | Acc. | Rob. | Acc. | Rob. | Acc. | Rob. | Acc. | Rob. | Acc. | Rob. | Acc. | Rob. | Acc. | Rob. |
| CLIP | 58.1 | 0.1 | 21.8 | 0.0 | 51.5 | 0.1 | 56.1 | 0.9 | 33.3 | 0.7 | 40.7 | 0.4 | 44.2 | 0.4 |
| TPT | 60.1 | 0.3 | 25.9 | 0.1 | 54.1 | 0.2 | 58.4 | 1.8 | 34.7 | 1.4 | 43.3 | 0.9 | 46.6 | 0.8 |
| Ensemble | 57.8 | 35.7 | 22.6 | 6.5 | 51.8 | 30.0 | 51.3 | 33.6 | 29.6 | 17.4 | 38.8 | 21.9 | 42.6 | 24.6 |
| TTC | 49.3 | 6.8 | 21.3 | 1.1 | 47.5 | 6.3 | 53.8 | 16.7 | 30.6 | 14.5 | 38.3 | 9.7 | 40.5 | 9.1 |
| DOC | 49.9 | 6.7 | 21.1 | 1.1 | 48.5 | 6.1 | 54.2 | 16.8 | 30.8 | 14.4 | 38.7 | 9.6 | 40.9 | 9.0 |
| R-TPT | 58.1 | 43.2 | 26.0 | 10.9 | 51.5 | 37.5 | 55.9 | 39.8 | 32.4 | 21.3 | 41.5 | 27.4 | 44.8 | 30.5 |
| Ours | 60.0 | 48.6 | 26.9 | 14.0 | 53.7 | 41.7 | 56.3 | 46.7 | 33.2 | 25.4 | 42.5 | 32.0 | 46.0 | 35.3 |

*Table 3.* Comparison of clean accuracy (Acc.) and adversarial accuracy (Rob., $\epsilon = 1.0$) of various adaptation methods on 10 fine-grained datasets using CLIP-ResNet50 with **only 1 augmented view**, highlighting the challenge under extremely limited visual diversity.

| Method | SUN397 | | Food101 | | Caltech101 | | DTD | | Flower102 | | Pets | | UCF101 | | Aircraft | | EuroSAT | | Cars | | Average | |
|---|---|---|---|---|---|---|---|---|---|---|---|---|---|---|---|---|---|---|---|---|---|---|---|
| | Acc. | Rob. | Acc. | Rob. | Acc. | Rob. | Acc. | Rob. | Acc. | Rob. | Acc. | Rob. | Acc. | Rob. | Acc. | Rob. | Acc. | Rob. | Acc. | Rob. | Acc. | Rob. |
| CLIP | 58.8 | 0.1 | 73.9 | 0.0 | 85.7 | 5.2 | 40.4 | 2.1 | 61.7 | 0.0 | 83.6 | 0.1 | 58.9 | 0.1 | 15.7 | 0.0 | 23.7 | 0.0 | 55.8 | 0.0 | 55.8 | 0.8 |
| TPT | 59.6 | 0.3 | 74.5 | 0.1 | 86.1 | 7.9 | 41.1 | 4.1 | 63.5 | 0.1 | 83.6 | 0.3 | 59.8 | 0.4 | 15.3 | 0.1 | 25.3 | 0.1 | 56.6 | 0.0 | 56.5 | 1.3 |
| Ensemble | 56.3 | 1.9 | 68.8 | 0.9 | 83.5 | 17.6 | 38.3 | 7.0 | 58.9 | 0.5 | 81.8 | 1.8 | 54.7 | 2.0 | 15.2 | 0.1 | 21.6 | 0.2 | 51.8 | 0.1 | 53.1 | 3.2 |
| TTC | 55.1 | 8.9 | 70.5 | 7.2 | 82.0 | 47.4 | 38.6 | 15.2 | 55.8 | 7.5 | 76.0 | 10.7 | 52.7 | 21.6 | 12.1 | 0.7 | 23.9 | 14.2 | 52.6 | 4.0 | 51.9 | 13.7 |
| DOC | 55.8 | 9.0 | 70.7 | 7.5 | 81.7 | 46.7 | 38.5 | 15.5 | 56.1 | 7.2 | 75.3 | 10.3 | 53.2 | 21.0 | 12.5 | 0.8 | 24.4 | 14.3 | 52.9 | 4.0 | 52.1 | 13.6 |
| R-TPT | 56.9 | 10.5 | 68.7 | 9.9 | 82.1 | 23.6 | 39.6 | 11.2 | 59.2 | 9.3 | 79.6 | 14.8 | 54.8 | 9.2 | 14.9 | 2.2 | 22.8 | 3.7 | 51.5 | 6.3 | 53.0 | 10.1 |
| Ours | 57.4 | 29.6 | 69.1 | 27.9 | 83.6 | 52.3 | 38.5 | 21.9 | 58.7 | 28.0 | 80.8 | 45.2 | 56.6 | 25.7 | 14.8 | 5.3 | 22.2 | 7.8 | 51.1 | 14.0 | 53.3 | 25.8 |

*Table 4.* Comparison of adversarial accuracy on Caltech101 and DTD under various attack methods, including CW (Carlini & Wagner), DI (DI$^2$-FGSM), AA (AutoAttack), and White-box attacks.

| Method | Caltech101 | | | | DTD | | | |
|---|---|---|---|---|---|---|---|---|
| | CW | DI | AA | White | CW | DI | AA | White |
| Ensemble | 77.0 | 63.3 | 70.7 | 66.9 | 32.2 | 23.6 | 29.1 | 20.0 |
| TTC | 59.4 | 42.0 | 45.2 | 54.5 | 25.1 | 14.1 | 17.1 | 15.8 |
| DOC | 61.8 | 41.8 | 46.8 | 57.9 | 26.8 | 14.2 | 18.0 | 17.3 |
| R-TPT | 76.0 | 66.5 | 73.6 | 66.2 | 32.7 | 26.4 | 32.7 | 18.9 |
| Ours | 81.2 | 75.2 | 80.6 | 69.3 | 35.4 | 30.0 | 35.5 | 22.1 |

*Table 5.* Comparison of average test time per image and adversarial accuracy on the Caltech101 and DTD datasets. For each method, the number of augmented views is chosen by prioritizing robustness while still considering test-time efficiency.

| Method | # Aug. Views | Caltech101 | | DTD | |
|---|---|---|---|---|---|
| | | Time (s) | Rob. (%) | # Time (s) | Rob. (%) |
| CLIP | - | 0.43 | 5.2 | 0.40 | 2.1 |
| TPT | 63 | 1.45 | 7.0 | 1.44 | 4.8 |
| Ensemble | 63 | 1.38 | 74.8 | 1.37 | 29.5 |
| TTC | - | 0.49 | 47.4 | 0.51 | 15.2 |
| DOC | - | 0.55 | 46.7 | 0.56 | 15.5 |
| R-TPT | 63 | 1.67 | 79.8 | 1.58 | 33.5 |
| Ours | 15 | 0.74 | **81.0** | 0.75 | **34.6** |

narios, and varying numbers of views. All experiments are conducted on an NVIDIA A100 GPU. Additional details are provided in the Appendix C.

## 4.2. Experimental Results

**Fine-grained benchmarks.** Across 10 fine-grained datasets, SS-TPT achieves state-of-the-art performance in both clean accuracy and robustness, as shown in Tab. 1. SS-TPT attains an average robustness of 46.8%, surpassing the strongest baseline R-TPT by **+6.4** points. Furthermore, SS-TPT attains an average clean accuracy of 55.1%, which is the closest to the original CLIP performance among all adaptation methods, demonstrating its superior ability to preserve clean accuracy while enhancing robustness.

**ImageNet and OOD variants.** On ImageNet and its shifted variants, SS-TPT achieves state-of-the-art performance in both clean and robustness, as shown in Tab. 2. On ImageNet, our method attains 60.0% clean accuracy and 48.6% robust accuracy, outperforming the strongest baseline R-TPT by **+1.9** and **+5.4** points, respectively. These gains generalize to ImageNet-A, -V2, -R, and -S, demonstrating strong resilience to distribution shifts. Notably, on average across these benchmarks, SS-TPT achieves the highest clean

accuracy (**46.0%**) and robustness (**35.3%**), even surpassing the original CLIP performance (44.2%) in clean accuracy.

**Robustness under diverse attacks.** Beyond PGD (Madry et al., 2018), SS-TPT demonstrates strong robustness against diverse attacks such as CW (Carlini & Wagner, 2017), DI$^2$-FGSM (Xie et al., 2019), AutoAttack (Croce & Hein, 2020), and white-box attacks as shown in Tab. 4. For the white-box attack, we design a pipeline-aware PGD that maximizes cross-entropy on the prediction of SS-TPT. This white-box attacker differentiates end-to-end through the SS-TPT pipeline, employing BPDA-STE (Athalye et al., 2018; Bengio et al., 2013) to traverse non-differentiable transforms. Although the white-box setting is challenging, SS-TPT maintains state-of-the-art performance. The consistent superiority of SS-TPT across diverse attack types highlights its strong generalizability as a robust defense method. More attack details are provided in the Appendix C.

**Challenging single-view case.** When only one augmented view is available, a challenging setting due to the lack of visual diversity, SS-TPT still delivers notable performance, as shown in Tab. 3. The average robust accuracy reaches

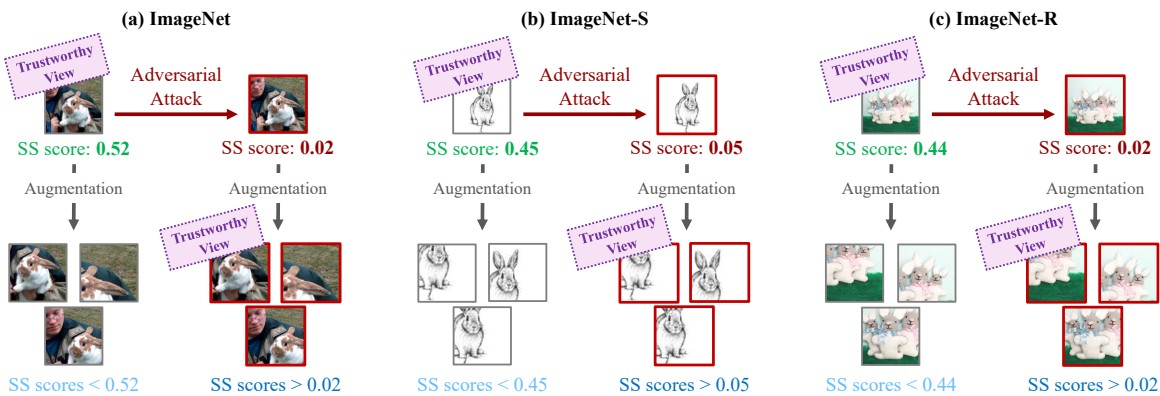

*Figure 5.* Average SS scores for original and adversarially attacked images on (a) ImageNet, (b) ImageNet-S, and (c) ImageNet-R. Scores are averaged over 500 random samples per dataset.

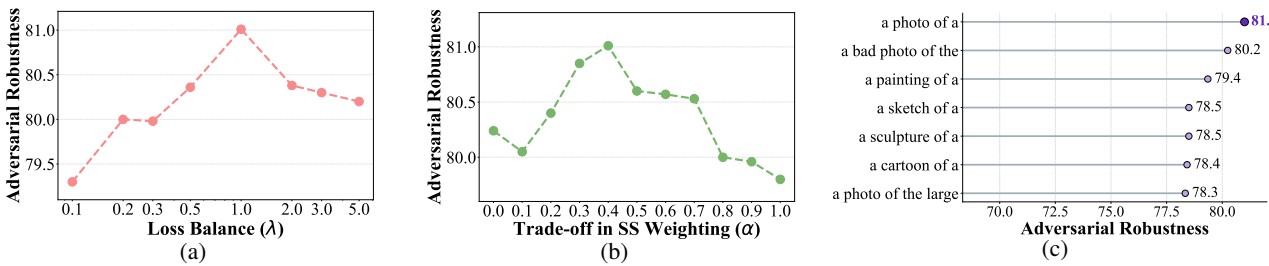

*Figure 6.* Ablation study on the Caltech101 dataset showing (a) the impact of the loss balance $\lambda$, (b) the trade-off between stability and suitability, and (c) robustness under variations in prompt templates.

25.8%, improving over R-TPT by **+15.7** points and over TTC by **+12.1** points. Clean accuracy is **53.3%**, establishing the highest performance among all adaptation methods.

**Test-time efficiency.** As shown in Tab. 5, SS-TPT achieves faster inference and stronger robustness. All methods are evaluated under an instance-level test-time adaptation protocol, where each test image is processed independently one at a time, without batching it with other test samples. With only 15 views, inference takes approximately 0.74–0.75 s per image, nearly **2 times faster** than R-TPT, Ensemble, and TPT. Meanwhile, robustness reaches **81.0%** on Caltech101 and **34.6%** on DTD, surpassing all competing methods.

**Interpretable guidance of SS scores.** We analyze the interpretability of SS-TPT by examining the average SS scores assigned to the original view and its adversarially attacked counterpart. As shown in Fig. 5, the SS scoring mechanism provides an intuitive criterion for identifying trustworthy views. For clean images, the unperturbed original view receives the highest SS score and therefore serves as the primary trustworthy view. Moreover, natural images from ImageNet obtain higher average scores than distribution-shifted variants, *e.g.*, sketches in ImageNet-S and artistic renditions in ImageNet-R. This indicates that the proposed SS score captures not only robustness-related reliability but also the naturalness of the source distribution.

Under adversarial attacks, the SS score of the original view drops sharply to near zero. In this case, SS-TPT shifts its emphasis from the corrupted original view to the augmented views, whose SS scores remain relatively higher because they are not directly optimized by the attack. This behavior demonstrates that SS-TPT can dynamically redirect adaptation and inference toward more trustworthy views depending on the input condition.

### 4.3. Ablation Study

**Loss balance.** We vary $\lambda$ to control the trade-off between point-wise entropy minimization and SS-guided consistency. Fig. 6a shows that adversarial robustness peaks at $\lambda=1$, where the consistency fully complements the entropy minimization. Performance remains stably high around this value, underscoring the benefit of aligning to stable and suitable views. However, as $\lambda$ decreases, robustness drops because the SS-guided alignment weakens, allowing corrupted views to exert greater influence during adaptation.

**Balancing stability and suitability.** We investigate how robustness is influenced by the coefficient $\alpha$, which controls the balance between stability and suitability in the SS score. As shown in Fig. 6b, the best performance is achieved at $\alpha=0.4$, which slightly favors stability over suitability. This suggests that under adversarial conditions, emphasizing in-

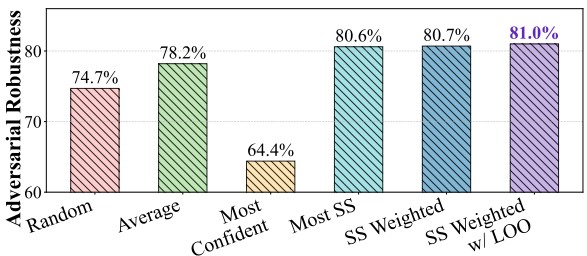

*Figure 7.* Comparison of adversarial accuracy under different reference selection strategies for $\mathcal{L}_{\text{scons}}$ on the Caltech101 dataset. The SS-based selection methods consistently outperform random, average, and confidence-based selection methods.

*Table 6.* Ablation study on 10 fine-grained classification datasets with all combinations of four components.

| Stability | Suitability | SS-Guided Consistency | SS-Weighted Prediction | Acc. | Rob. |
|:---:|:---:|:---:|:---:|:---:|:---:|
| ✗ | ✗ | ✗ | ✗ | 53.8 | 40.8 |
| ✓ | ✗ | ✓ | ✗ | 53.6 | 41.5 |
| ✓ | ✗ | ✗ | ✓ | 53.0 | 44.8 |
| ✓ | ✗ | ✓ | ✓ | 52.4 | 45.4 |
| ✗ | ✓ | ✓ | ✗ | 53.5 | 42.0 |
| ✗ | ✓ | ✗ | ✓ | 53.9 | 45.4 |
| ✗ | ✓ | ✓ | ✓ | 53.6 | 46.3 |
| ✓ | ✓ | ✓ | ✗ | 54.0 | 42.0 |
| ✓ | ✓ | ✗ | ✓ | 54.2 | 45.5 |
| ✓ | ✓ | ✓ | ✓ | **55.1** | **46.8** |

variance to stochastic perturbations is particularly beneficial. Crucially, leveraging both scores consistently outperforms relying on either one alone (*i.e.*, $\alpha=0$ or $\alpha=1$).

**Sensitivity analysis of prompt-template.** Fig. 6c presents an analysis of robustness under variations in prompt templates. We evaluate a diverse set of template styles, including descriptive, artistic, and deliberately degraded formulations, to assess whether the model relies on template-specific phrasing. Across all examined templates, adversarial robustness remains highly consistent and outperforms the state-of-the-art results reported in Tab. 1, indicating that SS-TPT generalizes well to diverse prompting conditions and does not depend on narrow linguistic structures of prompts.

**Reference selection for consistency.** To examine how different references affect the consistency loss, we compare: (1) *random*: using the prediction of a randomly chosen single view as a reference; (2) *average*: using the uniform average of predictions across all views; (3) *most-confident*: using the prediction of the lowest-entropy (most confident) single view; (4) *most-SS*: using the prediction of the single view with the highest SS score; (5) *SS-weighted*: using the SS-weighted average of predictions over views; and (6) *SS-weighted with LOO*: using an SS-weighted average formed in a leave-one-out manner for each target view as in Eq. (7). As shown in Fig. 7, *most-confident* performs worst, since low entropy does not guarantee a trustworthy view when perturbed views appear overly confident, and using them as a reference substantially harms robustness. In contrast, all SS-based strategies (*i.e.*, *most-SS*, *SS-weighted*, and *SS-weighted with LOO*) perform much better, demonstrating that combining stability and suitability successfully identifies trustworthy views. Among these, *SS-weighted with LOO* achieves the strongest results, as excluding the target view prevents self-reinforcement and allows each prediction to align with other trustworthy views.

**Component analysis.** Tab. 6 evaluates the contribution of stability, suitability, SS-guided consistency, and SS-weighted prediction. When combined with both consistency and weighted prediction, using either stability or suitabil-

ity alone already achieves strong performance, 45.4% and 46.3%, respectively. This shows that each score alone sufficiently captures view quality. While consistency alone is weak, its combination with a weighted prediction improves robustness by aligning each view with trustworthy references for more robust aggregation. The strongest results emerge when all proposed components are used, achieving the highest clean accuracy (55.1%) and robustness (46.8%), highlighting their synergy.

## 5. Conclusion

We introduce SS-TPT, a test-time defense for vision-language models that evaluates the quality of augmented views through stability and suitability scores. By leveraging qualified views to guide adaptation and inference, SS-TPT enforces consistency with trustworthy references and aggregates predictions for robust adaptation, even under limited visual diversity. Extensive experiments in various scenarios, including fine-grained benchmarks, ImageNet and its OOD variants, diverse attacks, and varying numbers of views, demonstrate that SS-TPT consistently achieves state-of-the-art robustness while preserving high clean accuracy. Furthermore, our method remains practical, as it enables effective adaptation even with a limited number of views. These results underscore the importance of view quality assessment in test-time adaptation and establish stability- and suitability-driven mechanisms as a promising foundation for effective adaptation of vision-language models.

**Limitations:** Our framework assumes stability and suitability are sufficient indicators of trustworthiness. A potential edge case arises if multiple corrupted augmented views happen to mutually cluster densely and behave consistently. In such rare scenarios, they could artificially yield high SS scores despite being semantically unreliable. To mitigate these boundary conditions, a natural next step for future work is to broaden the notion of view quality by incorporating complementary metrics, such as spatial cues that capture how consistently local regions behave across views.

## Acknowledgements

This research was supported by the Culture, Sports and Tourism R&D Program through the Korea Creative Content Agency grant funded by the Ministry of Culture, Sports and Tourism in 2026 (Project Name: Development of an Integrated Intelligence Platform Technology for Prohibited Substance Management, Project Number: RS-2026-25547939, Contribution Rate: 50%) and by the National Research Foundation of Korea (NRF) grant funded by the Korea government (MSIT) (No. RS-2026-25495369, Contribution Rate: 50%).

## Impact Statement

This paper introduces SS-TPT, a test-time prompt tuning defense that improves the robustness of vision-language models to distribution shifts and adversarial perturbations while mitigating the robustness-throughput trade-off common in multi-view test-time adaptation. If deployed responsibly, SS-TPT can increase reliability in real-world perception pipelines where inputs are noisy or manipulated, and can reduce unnecessary test-time computation costs by avoiding reliance on large numbers of views. By lowering energy consumption at inference time, SS-TPT also contributes to more environmentally sustainable deployment of vision-language systems. We encourage its use alongside careful evaluation, transparent reporting of limitations, and complementary safety practices to ensure responsible deployment.

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

*Table 7.* Datasets used in our evaluation. We report the number of classes and the size of the test split for each dataset. Our method performs test-time prompt tuning and is evaluated only on the test sets.

| Dataset | Description | # Classes | # Test |
|---|---|---|---|
| SUN397 | Scene images | 397 | 19,850 |
| Food101 | Food images | 101 | 30,300 |
| Caltech101 | Object category images | 101 | 2,465 |
| DTD | Texture images | 47 | 1,692 |
| Flower102 | Flower species images | 102 | 2,463 |
| Pets | Pet images (cats and dogs) | 37 | 3,669 |
| UCF101 | Human action images | 101 | 3,783 |
| Aircraft | Aircraft model images | 100 | 3,333 |
| EuroSAT | Satellite images | 10 | 8,100 |
| Cars | Car model images | 196 | 8,041 |
| ImageNet | Large-scale object images | 1,000 | 50,000 |
| ImageNet-A | Adversarial natural images | 200 | 7,500 |
| ImageNet-V2 | Re-collected ImageNet images | 1,000 | 10,000 |
| ImageNet-R | Artistic rendition images | 200 | 30,000 |
| ImageNet-S | Sketch images | 1,000 | 50,889 |

*Table 8.* Links to official implementations of baseline methods.

| Method | Official Implementation |
|---|---|
| TPT | github.com/azshue/TPT |
| MTA | github.com/MaxZanella/MTA |
| TTC | github.com/Sxing2/CLIP-Test-time-Counterattacks |
| DOC | github.com/bookman233/DOC |
| TAPT | github.com/xinwong/TAPT |
| R-TPT | github.com/TomSheng21/R-TPT |

## A. Future Work

Our method relies on two principled view-quality scores, *stability* and *suitability*, to identify which augmented views should drive test-time adaptation and inference. A natural next step is to broaden the notion of view quality beyond stability and suitability. For example, one could consider spatial cues that capture how consistently local regions behave across views. Exploring these aspects may lead to discovering additional complementary view-quality scores in future work.

## B. Datasets

The datasets cover a wide range of domains: general object categories (Caltech101, Food101), animal images (Pets), plant images (Flower102), vehicle images (Cars, Aircraft), texture images (DTD), remote sensing images (EuroSAT), human action images (UCF101), large-scale scene images (SUN397), and generalization and robustness benchmarks (ImageNet and its OOD variants). A detailed summary of dataset statistics is provided in Tab. 7.

## C. Further Experimental Details

**Baselines.** We compare our method against a diverse set of strong test-time adaptation and prompt-tuning baselines, including CLIP (Radford et al., 2021), TPT (Shu et al., 2022), MTA (Zanella & Ben Ayed, 2024), TTC (Xing et al., 2025), TAPT (Wang et al., 2025), and R-TPT (Sheng et al., 2025). We also include a simple multi-view ensemble baseline (Ensemble), which averages logits across $N$ views without any adaptation. For reproducibility and fairness, all baseline results are obtained by running the official implementations under the same evaluation protocol and using the standard dataset splits. The corresponding code repositories are pro-

vided in Tab. 8.

All methods are evaluated in the test-time instance-level adaptation regime. For each test example, the model performs on-the-fly adaptation on the same hardware, number of views, and augmentation stack. Importantly, because our method and all compared baselines are restricted to the same resources, they rely solely on pre-trained CLIP and standard data augmentations such as AugMix (Hendrycks et al., 2020), without making use of any external supervision, additional knowledge sources, or large-scale foundation models such as large language models. This ensures a fair comparison that isolates the effect of test-time adaptation itself.

For TAPT, the official release provides a loss that depends on pre-computed values available only for the CLIP-ViT-B/16 backbone. Consequently, we use the provided pre-computed quantities without modification and evaluate TAPT strictly on CLIP-ViT-B/16. In addition, TAPT can initialize prompts from adversarially trained prompts on ImageNet. To ensure a fair comparison focused purely on test-time adaptation and consistent with the no-external-supervision protocol, we disable any such initialization of prompts across all baselines.

**Augmentation for stability.** To compute the stability score, our augmentation ranges are adapted from common practices in data augmentation for robustness (*e.g.*, AugMix (Hendrycks et al., 2020)), but tuned to be milder to preserve semantic labels. Concretely, we apply random affine transformations (rotation $\pm 5°$, translation $4\%$, scale $\pm 8\%$), color jitter (brightness/contrast/saturation $\pm 0.08$, hue $\pm 0.02$), Gaussian blur ($\sigma \in [0.5, 1.2]$), and additive Gaussian noise ($\sigma \leq 0.02$). Each transformation is applied with a fixed probability: affine ($p{=}1.0$) and all others ($p{=}0.5$), yielding mild yet diverse perturbations without altering the underlying semantics.

**Adversarial attacks.** We adopt the adversarial attack configuration of (Sheng et al., 2025) to ensure strict comparability. All attacks operate on image tensors in $[0, 1]$ where the perturbation budget is set to $\epsilon/255$. Random start is enabled and each attack uses one restart. To ensure a comparable perturbation strength across different backbones and attack scenarios, the hyperparameters are adjusted accordingly.

*Table 9.* Ablation study on the Caltech101 dataset.

(a) Different methods for suitability scores.

| Method | Rob. |
|---|---|
| Mean Similarity | 80.6 |
| Inverse Mean Distance | **81.0** |

(b) Divergence measures for stability scores.

| Divergence Measure | Rob. |
|---|---|
| Symmetric KL Divergence | 80.8 |
| Jensen-Shannon Divergence | **81.0** |

(c) Impact of individual augmentations on stability scores via drop- and add-one ablation.

| Augmentation Configuration | Rob. |
|---|---|
| Base Config. | **81.0** |
| – Affine Transformation | 77.2 |
| – Color Jitter | 79.9 |
| – Gaussian Blur | 79.7 |
| – Gaussian Noise | 80.2 |
| + Horizontal Flip | 80.5 |

(d) Sampling strategy for $\mathcal{L}_{\text{scons}}$.

| Sampling Strategy | Rob. |
|---|---|
| All Views | 79.1 |
| Selected Views | **81.0** |

(e) Normalization for SS scores.

| Normalization for Scores | Rob. |
|---|---|
| Z-score Norm. | 78.9 |
| Min-Max Norm. | **81.0** |

- **PGD** ($L_\infty$): For CLIP-ResNet50, $\epsilon = 1$ and 7 steps with step size $\epsilon/4 = 0.25$. For CLIP-ViT, $\epsilon = 4$ and 100 steps with step size $\epsilon/4 = 1$.

- **DI$^2$-FGSM** ($L_\infty$): Same $\epsilon$ and step size as PGD with step count doubled. Other diversity-transform parameters follow library defaults.

- **AutoAttack** ($L_\infty$): Standard version with the same $\epsilon$ as PGD. All internal thresholds and presets are left at defaults.

- **CW** ($L_2$): $c = 1.0$, $\kappa = 0$, 500 iterations, learning rate 0.01.

- **White-box attack:** We design an SS-TPT pipeline-aware PGD that differentiates end-to-end through our SS-TPT pipeline, employing BPDA-STE (Athalye et al., 2018; Bengio et al., 2013) for non-differentiable augmentations. Given a test image, we generate a set of views $\{x_n\}_{n=0}^N$, consisting of the original view and its $N$ augmented views. With the final prediction of SS-TPT in Eq. (9), the attacker maximizes

$$\mathcal{L}_{\text{white}}(x_0, y) = \text{CE}(\widetilde{p}(\cdot \mid x_0), y), \qquad (10)$$

where CE denotes the standard cross-entropy loss and $y$ is the ground-truth label. We set $N = 15$ (same as $N$ in the defense) and use Expectation over Transformation (EOT) with 10 samples per PGD step, while keeping the PGD hyperparameters (*e.g.*, $\epsilon$, step size, number of steps, and random start) the same as in our standard PGD setting.

## D. More Experiments

**Memory consumption.** We measure peak GPU memory consumption (MB) on Caltech101 under identical hardware and evaluation protocols with $N = 15$, as shown in Tab. 10. Lightweight methods such as TTC and DOC remain under ~1 GB, as they avoid per-instance gradient updates. In contrast, prompt tuning-based methods (TPT, R-TPT, and our SS-TPT) require ~2.3 GB, reflecting the additional buffers

*Table 10.* Comparison of peak GPU memory using CLIP-ResNet50 on Caltech101.

| Method | Peak Memory (MB) |
|---|---|
| CLIP | 892.8 |
| Ensemble | 941.4 |
| TTC | 939.1 |
| DOC | 935.1 |
| TPT | 2354.9 |
| TAPT | 3649.2 |
| R-TPT | 2354.9 |
| Ours | 2354.9 |

for the optimizer state needed for the inner gradient update at test time. Despite their higher memory consumption, prompt-tuning approaches achieve substantially stronger adversarial robustness than lightweight alternatives, as shown in Tab. 13. In practice, memory can be reduced by lowering the number of views $N$.

**Further ablation studies** Further ablation studies for SS scores are included in Tab. 9. The results reveal several findings. (i) Across suitability metrics, inverse mean distance slightly outperforms mean similarity. We attribute this to cosine-based mean similarity being purely angular, which is insensitive to feature norms and local sample density, whereas inverse mean distance explicitly captures local density. (ii) For stability, Jensen-Shannon divergence is a better choice than symmetric KL. (iii) Among weak augmentations, affine transforms contribute most, while color jitter, blur, and light noise provide smaller but consistent gains. Notably, adding horizontal flip slightly degrades robustness because flip can alter the semantic meaning for certain categories (*e.g.* text, asymmetric objects). (iv) Using a selected subset for the SS-guided consistency loss is superior to using all views. This is because enforcing consistency across every augmentation is inefficient and may propagate noise from less informative views, which aligns with prior findings (Shu et al., 2022; Sheng et al., 2025). (v) Finally, min-max normalization is clearly better than z-score normalization for combining SS scores. Since min-max normalization guarantees bounded values on a common scale, it ensures comparability across scores and avoids distortions that arise from differences in their underlying distributions.

*Table 11.* Comparison of clean accuracy (Acc.) and adversarial accuracy (Rob., $\epsilon$=1.0) of **training-time defense methods** and ours with 63 augmented views on 8 fine-grained classification datasets using CLIP-ResNet50. For training-time defense methods, we use the results reported in (Sheng et al., 2025).

| Method | Caltech101 | | DTD | | Flower102 | | Pets | | UCF101 | | Aircraft | | EuroSAT | | Cars | | Average | |
|---|---|---|---|---|---|---|---|---|---|---|---|---|---|---|---|---|---|---|
| | Acc. | Rob. | Acc. | Rob. | Acc. | Rob. | Acc. | Rob. | Acc. | Rob. | Acc. | Rob. | Acc. | Rob. | Acc. | Rob. | Acc. | Rob. |
| CLIP (Radford et al., 2021) | 85.9 | 2.6 | 40.4 | 0.8 | 61.7 | 0.0 | 83.6 | 0.0 | 59.0 | 0.0 | 15.7 | 0.0 | 23.7 | 0.0 | 55.7 | 0.0 | 53.2 | 0.4 |
| TeCoA (Mao et al., 2023) | 78.3 | 78.3 | 26.2 | 26.0 | 33.5 | 33.4 | 76.0 | 75.8 | 38.4 | 38.1 | 5.8 | 5.8 | 16.5 | 16.6 | 22.4 | 22.3 | 37.1 | 37.0 |
| APT (Li et al., 2024) | 2.9 | 1.7 | 16.6 | 7.9 | 2.6 | 1.1 | 31.9 | 3.8 | 11.2 | 0.9 | 0.9 | 0.9 | 17.0 | 4.0 | 8.5 | 0.6 | 11.4 | 2.6 |
| APT+TeCoA (Li et al., 2024) | 82.8 | 82.8 | 39.2 | 39.0 | 42.7 | 42.6 | 79.3 | 79.0 | 51.5 | 51.4 | 9.9 | 9.7 | 32.9 | 32.9 | 33.9 | 33.6 | 46.5 | 46.4 |
| Ours (test-time) | 86.4 | 81.2 | 39.8 | 33.8 | 59.4 | 51.3 | 83.0 | 73.7 | 58.3 | 51.0 | 18.2 | 13.6 | 22.6 | 19.0 | 56.4 | 42.5 | 53.0 | 45.8 |

*Table 12.* Comparison of clean accuracy (Acc.) and adversarial accuracy (Rob., $\epsilon = 4.0$) of various adaptation methods on 10 fine-grained datasets using TeCoA pre-trained CLIP-ViT-B/32 with 15 augmented views.

| Method | SUN397 | | Food101 | | Caltech101 | | DTD | | Flower102 | | Pets | | UCF101 | | Aircraft | | EuroSAT | | Cars | | Average | |
|---|---|---|---|---|---|---|---|---|---|---|---|---|---|---|---|---|---|---|---|---|---|---|
| | Acc. | Rob. | Acc. | Rob. | Acc. | Rob. | Acc. | Rob. | Acc. | Rob. | Acc. | Rob. | Acc. | Rob. | Acc. | Rob. | Acc. | Rob. | Acc. | Rob. | Acc. | Rob. |
| CLIP | 33.2 | 6.6 | 22.4 | 3.3 | 78.9 | 45.9 | 24.5 | 11.8 | 30.8 | 10.0 | 66.9 | 18.5 | 34.6 | 7.2 | 6.6 | 0.7 | 14.5 | 10.8 | 10.2 | 1.1 | 32.3 | 11.6 |
| TPT | 32.7 | 8.9 | 21.1 | 4.6 | 78.9 | 51.7 | 24.9 | 14.3 | 27.7 | 12.5 | 64.4 | 27.0 | 34.9 | 9.9 | 6.6 | 1.2 | 12.4 | 11.1 | 9.6 | 2.0 | 31.3 | 14.3 |
| Ensemble | 26.4 | 12.8 | 14.9 | 6.2 | 72.8 | 56.0 | 23.3 | 16.0 | 26.6 | 15.9 | 59.9 | 37.9 | 27.0 | 14.1 | 4.4 | 1.9 | 12.7 | 10.6 | 5.8 | 2.7 | 27.4 | 17.4 |
| R-TPT | 27.6 | 13.1 | 14.9 | 6.9 | 73.9 | 55.9 | 24.4 | 17.0 | 26.0 | 15.5 | 60.4 | 37.0 | 28.2 | 14.5 | 5.4 | 2.2 | 11.9 | 10.0 | 6.1 | 2.9 | 27.9 | 17.5 |
| Ours | 27.4 | 13.3 | 15.7 | 7.0 | 73.2 | 56.0 | 25.0 | 17.7 | 26.3 | 16.2 | 60.6 | 38.9 | 28.3 | 14.9 | 5.5 | 2.6 | 12.7 | 11.1 | 6.4 | 3.1 | 28.1 | 18.1 |

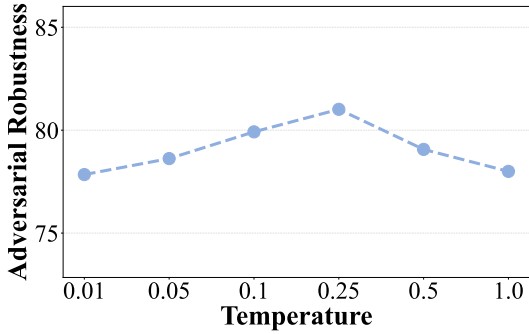

*Figure 8.* Ablation study on the Caltech101 dataset showing the impact of the temperature.

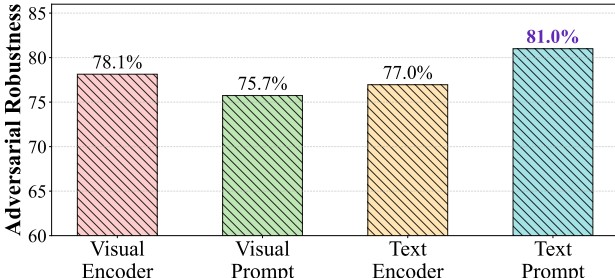

*Figure 9.* Robustness across different optimized parameter configurations on the Caltech101 dataset. The results consistently demonstrate state-of-the-art performance, indicating that our approach remains effective not only for text prompt tuning but also for other optimization schemes, such as encoder tuning and visual prompt tuning.

**Temperature sensitivity.** We analyze the sensitivity of SS-TPT to the temperature parameter $\tau_w$ used in the SS-weighted softmax. As shown in Fig. 8, adversarial robustness on Caltech101 reaches its highest value at $\tau_w = 0.25$. Importantly, the performance remains stable across a broad range of temperatures, from very sharp weighting to smoother weighting. This indicates that SS-TPT is not overly sensitive to the precise choice of $\tau_w$, and that the proposed stability and suitability scoring provides reliable view-quality estimates under different weighting sharpness levels.

**Sensitivity analysis of different optimized parameter configurations.** Fig. 9 further investigates robustness under varying optimized parameter configurations. The results show that performance varies slightly across parameter configurations. Notably, these results still achieve state-of-the-art performance, consistently surpassing the compared methods as shown in Tab. 1. This behavior indicates that SS-TPT remains effective not only for text prompt tuning but also across other optimization schemes such as encoder tuning and visual prompt tuning.

**Comparison to training-time defenses.** Tab. 11 shows that our test-time method achieves robustness competitive with strong training-time defenses, although our method requires no labels and is entirely training-free. While our SS-TPT is marginally lower in robust accuracy compared to APT+TeCoA (45.8% vs. 46.4%), SS-TPT surpasses APT+TeCoA in clean accuracy by a large margin (53.0% vs. 46.5%). This highlights that test-time, label-free adaptation can rival fully trained defenses while being more flexible to deploy with its test-time design.

**Results with TeCoA pretraining.** As shown in Tab. 12, using TeCoA-pretrained CLIP-ViT-B/32, our method attains the highest average robust accuracy (18.1%), outperforming other test-time baselines across diverse fine-grained datasets. These gains indicate that SS-guided adaptation is complementary to adversarial pretraining and remains effective even with adversarially robust backbones.

**Generality across architectures.** To further validate the

*Table 13.* Comparison of clean accuracy (Acc.) and adversarial accuracy (Rob., $\epsilon = 4.0$) of various adaptation methods on 10 fine-grained classification datasets using CLIP-ViT-B/16 with 15 augmented views.

| Method | SUN397 | | Food101 | | Caltech101 | | DTD | | Flower102 | | Pets | | UCF101 | | Aircraft | | EuroSAT | | Cars | | Average | |
|---|---|---|---|---|---|---|---|---|---|---|---|---|---|---|---|---|---|---|---|---|---|---|
| | Acc. | Rob. | Acc. | Rob. | Acc. | Rob. | Acc. | Rob. | Acc. | Rob. | Acc. | Rob. | Acc. | Rob. | Acc. | Rob. | Acc. | Rob. | Acc. | Rob. | Acc. | Rob. |
| CLIP | 62.6 | 0.0 | 83.6 | 0.0 | 93.3 | 0.0 | 44.4 | 0.0 | 67.4 | 0.0 | 88.3 | 0.0 | 65.2 | 0.0 | 24.0 | 0.0 | 42.2 | 0.0 | 65.5 | 0.0 | 63.7 | 0.0 |
| TPT | 64.6 | 0.0 | 84.1 | 0.0 | 93.6 | 0.0 | 46.2 | 0.0 | 67.4 | 0.0 | 86.0 | 0.0 | 67.5 | 0.0 | 21.6 | 0.0 | 42.9 | 0.0 | 66.3 | 0.0 | 64.0 | 0.0 |
| Ensemble | 64.1 | 23.3 | 81.2 | 21.2 | 92.3 | 57.8 | 44.0 | 17.4 | 65.6 | 17.1 | 86.4 | 28.9 | 63.0 | 19.5 | 23.3 | 4.1 | 29.0 | 0.3 | 64.8 | 10.3 | 61.4 | 20.0 |
| TTC | 20.8 | 3.6 | 82.5 | 5.7 | 92.4 | 9.9 | 41.8 | 5.5 | 65.0 | 7.5 | 81.1 | 10.8 | 64.5 | 1.8 | 20.8 | 0.3 | 53.4 | 0.2 | 60.9 | 3.3 | 58.3 | 4.9 |
| DOC | 58.1 | 7.2 | 82.8 | 7.1 | 92.4 | 6.5 | 42.0 | 12.2 | 64.6 | 9.7 | 75.7 | 10.2 | 64.1 | 0.8 | 21.2 | 0.2 | 39.8 | 0.0 | 62.0 | 2.2 | 60.3 | 5.6 |
| TAPT | 64.1 | 13.7 | 86.7 | 4.9 | 91.6 | 38.3 | 43.6 | 12.6 | 63.1 | 9.1 | 87.3 | 14.5 | 65.9 | 7.1 | 20.1 | 2.9 | 42.8 | 6.9 | 63.2 | 4.4 | 62.8 | 11.4 |
| R-TPT | 64.1 | 43.9 | 82.2 | 46.4 | 92.6 | 78.5 | 44.9 | 26.7 | 66.7 | 44.7 | 85.8 | 60.7 | 65.5 | 40.7 | 21.7 | 11.7 | 34.1 | 7.3 | 64.5 | 30.6 | 62.2 | 39.1 |
| Ours | 64.8 | 45.6 | 82.6 | 48.3 | 93.0 | 81.5 | 45.0 | 31.9 | 67.0 | 45.4 | 86.2 | 61.1 | 66.0 | 43.1 | 22.5 | 12.8 | 34.1 | 7.6 | 65.1 | 33.6 | 62.6 | 41.1 |

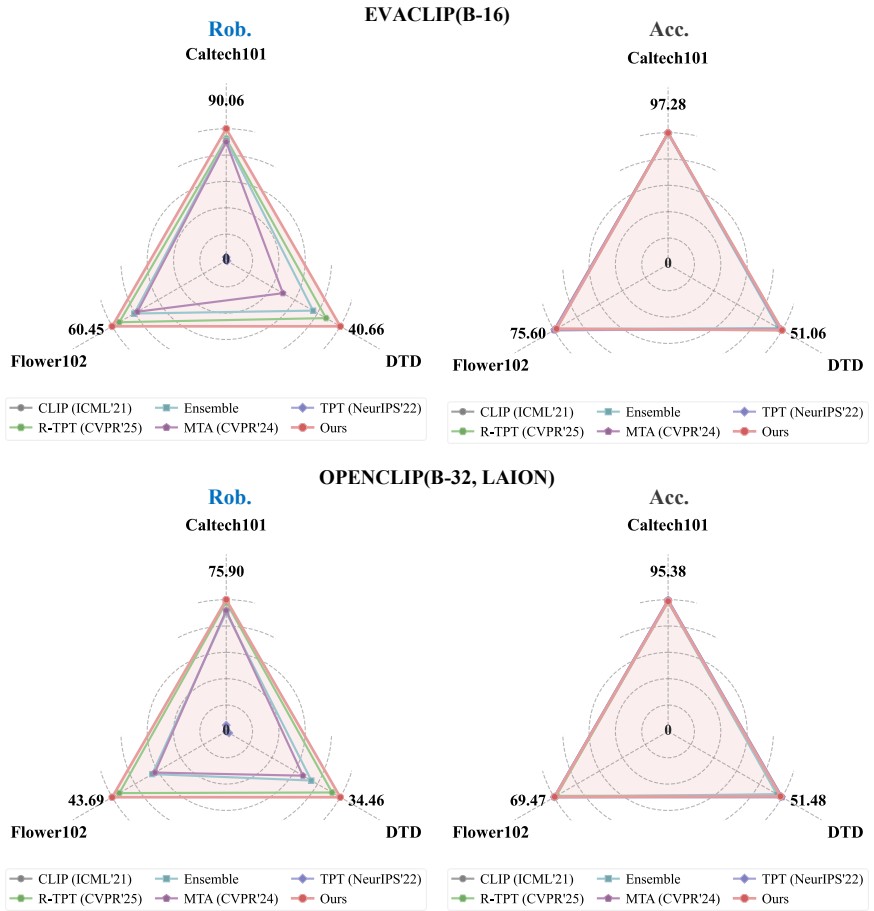

*Figure 10.* Comparison of clean accuracy (Acc.) and adversarial accuracy (Rob.) under PGD attack ($\epsilon = 4/255$, step 10) across two architectures: EVA02-B-16 and OpenCLIP(VIT-B-32, LAION pretrained). Results are evaluated on three fine-grained datasets.

generality of SS-TPT, we conduct additional experiments on different vision-language model architectures.

Tab. 13 shows that SS-TPT achieves the highest adversarial accuracy on CLIP-ViT-B/16 model. Notably, under strong perturbations ($\epsilon = 4.0$), standard methods such as zero-shot CLIP and TPT fail (0.0% robustness), whereas SS-TPT maintains substantial robustness (41.1%). These results demonstrate that the effectiveness of our stability and suitability-guided adaptation generalizes well to the large architecture.

As shown in Fig. 10, we evaluate SS-TPT on EVA02-B-16 and OpenCLIP ViT-B-32 pretrained on LAION across three fine-grained datasets under PGD attack. SS-TPT consistently improves adversarial robustness over adaptation baselines while preserving clean accuracy across both architectures. These results indicate that the proposed stability–suitability scoring mechanism is not tied to a specific CLIP backbone or pre-training distribution, but can effectively identify trustworthy views across diverse model architectures.

