# OpenReview forum: "SS-TPT: Stability and Suitability-Guided Test-Time Prompt Tuning for Adversarially Robust Vision-Language Models"
_ICML.cc/2026/Conference — ICML 2026 regular_

### Official Review · Reviewer_vveA · 2026-02-27

**Soundness:** 3
**Presentation:** 3
**Significance:** 3
**Originality:** 3
**Overall Recommendation:** 4
**Confidence:** 4

**Summary:**

This paper addresses the robustness-throughput trade-off in defending VLMs against adversarial attacks at test time. Existing test-time prompt tuning methods require numerous augmented views for robustness, causing high inference latency. The proposed SS-TPT shifts focus from view quantity to quality by scoring each view on Stability and Suitability. These scores guide both a consistency loss for prompt tuning and a weighted prediction aggregation at inference. Experiments across multiple benchmarks and attack settings show that SS-TPT maintains strong robustness with significantly fewer views, achieving a superior robustness-throughput trade-off.

**Compliance With Llm Reviewing Policy:**

Affirmed.

**Final Justification:**

I thank the authors for their thorough and thoughtful response. In the rebuttal, they directly addressed my major concerns. I appreciate these clarifications and will therefore maintain my positive score.

**Key Questions For Authors:**

1. The paper focuses on improving the robustness of CLIP. While this is a significant contribution, the experiments are all conducted on CLIP model only, without results on other CLIP-like models, e.g., OpenCLIP, EVA-CLIP, which may limit the contribution of proposed methods.

2. It is unclear whether the SS-TPT is applied only at the embedding layer (shallow, as in CoOp) or across multiple transformer layers (deep, as in layers 0-9). If SS-TPT relies solely on shallow, textual prompts, its limited parameterization may constrain performance. Indeed, prior work on both train-time defenses (e.g., MaPLe) and test-time defenses (e.g., TAPT) has shown that deep, multi-modal prompt tuning yields notably stronger robustness than shallow, single-modal alternatives.

3. Adaptive Attacks. It is recommended to evaluate against more rigorous, systematic adaptive attacks specifically tailored to bypass the SS weighting mechanism.

**Limitations:**

Reference Weaknesses and Questions

**Strengths And Weaknesses:**

Strengths

1. The manuscript is well-organized and readable.

2. The experimental setup is described in a detailed and thorough manner.

Weaknesses

1. The SS‑TPT computational cost may be very high.

---

> ### Author Rebuttal · Authors · 2026-03-30
>
> We thank the reviewer for the valuable feedback.
>
> >  **Q1.** Computational cost
>
> Our Stability and Suitability (SS) strategy is highly efficient, with SS-TPT providing a particularly effective trade-off between robustness and efficiency (Fig. 1). Furthermore, it substantially reduces the total computational burden for the following reasons:
> **Selective Computation:** The stability calculation does not scale linearly with the total number of views. By filtering for a low-entropy subset $ \mathcal{B} $, stability is evaluated only on confident views, strictly limiting overhead.
>
> **Superior Performance with Fewer Views:** Guided by SS scores, SS-TPT achieves superior robustness (81.0%) with only 16 views, whereas the previous state-of-the-art R-TPT requires 64 views for 79.8%. Since augmentation dominates inference time, fewer views can cut latency ($ 1.67\text{s} \to 0.74\text{s} $) and FLOPs ($ 2081.4\text{G} \to 693.8\text{G} $), as shown in **Fig. A**.
>
> * Fig. A: https://anonymous.4open.science/r/qk7/FigA.png
>
> >  **Q2.** More VLM models
>
> To further validate the generality of our SS-TPT, we conduct additional experiments on various architectures. As shown in **Fig. E**, we evaluate SS-TPT on **EVA02-B-16** and **OpenCLIP (ViT-B-32, LAION-pretrained)** across multiple fine-grained datasets. SS-TPT consistently maintains superior performance in adversarial robustness (Rob.) on these models, while preserving clean accuracy (Acc.) This confirms that our scoring mechanism effectively identifies trustworthy views regardless of the specific pre-training data or architecture.
>
> The effectiveness of SS-TPT is further supported by extensive evaluations in the main manuscript across a wide range of backbones, including CNN-based **ResNet-50** (Tab. 1), Transformer-based **ViT-B/16** (Tab. 9), and even an adversarially robust **TeCoA pre-trained CLIP-ViT-B/32** (Tab. 13).
>
> In all examined settings, SS-TPT consistently achieved the highest performance among all adaptation baselines, demonstrating its strong generality.
>
> * Fig. E: https://anonymous.4open.science/r/qk7/FigE.png
>
> >  **Q3.** Scope of parameterization
>
> For fair comparisons with TPT-based methods, we use shallow textual prompt tuning with 4 learnable context tokens. However, our SS guidance remains highly effective across various tuning schemes.
>
> **Robustness across Different Tuning Targets:** As shown in **Fig. F**, SS-TPT consistently achieves state-of-the-art performance compared to R-TPT (69.3%), regardless of the tuning target: * **Visual Encoder Tuning**: 78.1%. * **Visual Prompt Tuning**: 75.7%. * **Text Encoder Tuning**: 77.0%. * **Text Prompt Tuning**: 81.0%.
> These results confirm that our SS guidance is the primary driver of robustness, not the specific parameter space.
>
> **View Quality vs. Parameterization Depth:** While deep parameterization-based works like TAPT emphasize the depth of parameterization, SS-TPT achieves significantly stronger robustness by prioritizing **view quality over parameter volume**. In Tab. 9, SS-TPT achieves **41.1%** robust accuracy, far outperforming TAPT’s **11.4%**. Furthermore, deep tuning is computationally heavy. TAPT requires a peak memory of **3649.2 MB**, whereas SS-TPT operates efficiently at **2354.9 MB** (Tab. 11). Our findings suggest that the **SS-guided mechanism** provides a more critical and efficient advantage for test-time defense than simply increasing the depth of prompt parameterization.
>
> * Fig. F: https://anonymous.4open.science/r/qk7/FigF.png
>
> >  **Q4.** Tailored adaptive attacks targeting the SS mechanism
>
> **Adaptive Attack Settings:** To test if the SS score can be hijacked, we first design a fully pipeline-aware white-box attack that maximizes the cross-entropy of our prediction (More details in Eq. 10):
>
> $$
> \mathcal{L}_{\text{adaptive}}(x_0,y) = \mathrm{CE}(\widetilde{p}(\cdot\,|\,x_0),y).
> $$
>
> We further augmented this with an **SS-Distortion** loss that forces the model to over-weight the corrupted original view ($w_0$):
>
> $$
> \mathcal{L}_{\text{adaptive+distortSS}}(x_0,y) = \mathrm{CE}(\widetilde{p}(\cdot\,|\,x_0),y) - \gamma \log(1 - w_0).
> $$
>
> For the stochastic weak augmentations used in the stability score, we apply EOT with 10 Monte Carlo samples at each PGD step using the same $N=15$ views as in the defense, so the attacker optimizes the expected loss over the same randomness rather than a single sampled augmentation. We further evaluate stronger adaptive attacks by increasing both the budget (eps) and the number of steps.
>
> **Resilience to Adaptive Attacks:** As shown in **Tab. B**, this tailored attack succeeds in reducing robustness. Under such heavy distortions, the image semantics are severely degraded, causing all other defense baselines to break down. Crucially, even when explicitly targeted, **SS-TPT consistently maintains the highest adversarial accuracy among all baselines, demonstrating its resilience.**
>
> * Tab. B: https://anonymous.4open.science/r/qk7/TabB.png

---

> > ### Author Rebuttal · Reviewer_vveA · 2026-04-02
> >
> > I thank the authors for their thorough and thoughtful response. In the rebuttal, they directly addressed my major concerns. I appreciate these clarifications and will therefore maintain my positive score.

---

> > > ### Author Response · Authors · 2026-04-03
> > >
> > > We truly appreciate your time in reviewing our rebuttal and your decision to maintain a positive score. It is highly encouraging to know that our additional analyses effectively cleared up your main concerns. Your insightful feedback enabled us to extensively validate SS-TPT across diverse VLMs, tuning depths, and targeted adaptive attacks. We are fully committed to incorporating all these newly conducted experiments into the final version of our paper. Thank you once again for your invaluable guidance and support.

---

### Official Review · Reviewer_VVEX · 2026-03-09

**Soundness:** 3
**Presentation:** 4
**Significance:** 3
**Originality:** 2
**Overall Recommendation:** 5
**Confidence:** 4

**Summary:**

This paper introduces Stability and Suitability-guided Test-time Prompt Tuning (SS-TPT), a test-time adaptation (TTA) defense designed to improve the adversarial robustness of Vision-Language Models (VLMs) like CLIP. SS-TPT evaluates the quality of each augmented view using stability (the invariance of a view's prediction to weak stochastic augmentations) and suitability (the density of the view in the feature space relative to other views). These scores are combined to create weights that guide both the adaptation phase and the inference phase. Through experiments across multiple datasets and various attack types, the authors demonstrate that SS-TPT achieves state-of-the-art adversarial robustness and clean accuracy while significantly improving throughput by requiring fewer augmented views.

**Compliance With Llm Reviewing Policy:**

Affirmed.

**Key Questions For Authors:**

1. In your white-box attack pipeline, how was the Expectation over Transformation (EOT) scaled specifically to account for the stochastic augmentations $\mathcal{A}'$ used to compute the internal stability score? Could an adaptive attacker theoretically bypass the defense by crafting perturbations that artificially inflate the stability score of the corrupted view?
2. You noted that hyperparameters such as $\alpha=0.4$ and $\lambda=1$ were kept constant across all datasets. Did you observe any specific datasets (e.g., highly stylized sets) where dynamically tuning the stability-suitability trade-off would have yielded a different optimal $\alpha$?
3. Appendix D notes that SS-TPT requires ~2.3 GB of peak memory compared to ~0.9 GB for lightweight methods like TTC. Since you emphasize the robustness-throughput trade-off, could you provide insights into the memory-robustness trade-off? Specifically, how does scaling the number of views $N$ dynamically impact peak memory consumption during the backpropagation step?

**Limitations:**

No, the authors have not adequately discussed the limitations in the main text.

**Strengths And Weaknesses:**

***Strength***
+ **Soundness**: The methodology is technically very sound and addresses a clear vulnerability in current multi-view TTA defenses. The mathematical formulation of stability using Jensen-Shannon divergence and suitability using inverse mean distance is logical and empirically validated in the ablations. The experimental design is thorough, testing across 10 fine-grained datasets, ImageNet, and 4 OOD variants , as well as against multiple attack paradigms.
+ **Presentation**: The paper is readable, well-structured, and clearly contextualizes its contributions against concurrent literature. For example, Figure 3 provides an intuitive visual summary of the entire SS-TPT pipeline, making the mathematical formulations much easier to understand.
+ **Significance**: This paper solves a highly relevant and practical problem: deploying robust VLMs in the real world without crippling inference speeds. The finding that SS-TPT can maintain strong robustness even in the extremely challenging 1-view regime is highly significant for latency-critical applications.
+ **Originality**: Although TPT and entropy minimization are standard, explicitly quantifying the trustworthiness of individual views at test time to filter out adversarial noise is a creative and highly effective angle. The combination of stability and suitability provides a novel perspective on filtering views during test-time augmentation.

***Weakness***
+ **Soundness**: The white-box attack evaluation relies on BPDA-STE to bypass non-differentiable transformations. While this is a standard approximation, adaptive attackers with full knowledge of the SS scoring mechanism might find edge cases to explicitly minimize the suitability score of clean views. Furthermore, the hyperparameters ($\alpha=0.4, \lambda=1$) are kept strictly fixed across all datasets and attack scenarios. It leaves open the question of whether the method is operating sub-optimally in specific domain shifts.
+ **Significance**: While 0.74s per image is much faster than R-TPT, it is still nearly double the latency of vanilla CLIP (0.43s). For high-frequency real-time systems, this overhead may still limit the deployment.
+ **Originality**: The underlying framework relies heavily on the architecture of previous methods like TPT and R-TPT.

---

> ### Author Rebuttal · Authors · 2026-03-30
>
> We thank the reviewer for the constructive feedback.
>
> >  **Q1.** EOT scaling & Stronger adaptive settings
>
> Due to space constraints, please refer to our response to **Reviewer vveA (Q4)** and **Tab. B**. We confirm the consistent superiority of SS-TPT over baselines under these harsher adaptive attacks.
>
> >  **Q2.** Fixed hyperparameters across datasets
>
> We agree that the optimal hyperparameters may vary across datasets. However, we intentionally use a single fixed hyperparameter configuration across diverse settings to demonstrate the practicality and generality of SS-TPT. We tuned the hyperparameters on Caltech and kept them fixed throughout all experiments. While this choice may not be optimal for every dataset, it was sufficient to achieve SOTA results across diverse fine-grained datasets, the general ImageNet dataset, distribution-shift benchmarks, attack scenarios, and varying numbers of views.
>
> That said, we agree with the reviewer that more specialized tuning could further improve performance in certain domains. For example, on Pets dataset, robustness improves from $72.8$% to $73.2$% with $\alpha = 0.8$. Thus, although dataset-specific tuning was beyond the scope of this paper, our results suggest that SS-TPT has clear potential for additional gains when adapted to individual domains. We will include this discussion on dataset-specific hyperparameter tuning in the revised manuscript.
>
> >  **Q3.** Still substantial latency compared to CLIP
>
> While SS-TPT is slower than vanilla CLIP ($0.74$ s vs. $0.43$ s per image), vanilla CLIP exhibits only near-zero adversarial robustness (Tab. 1). This makes **CLIP’s speed advantage insufficient in safety-critical settings** where intentional attacks can cause severe failures, such as autonomous driving, medical decision support, or security applications. In these scenarios, robust defense is essential, and among the defense methods, SS-TPT offers a particularly effective trade-off between robustness and efficiency (Fig. 1).
>
> More concretely, compared to the recent state-of-the-art R-TPT, **Fig. A** shows that SS-TPT achieves higher robustness ($81.0$% vs. $79.8$%) with much lower total latency ($0.74$ s vs. $1.67$ s) and significantly fewer FLOPs ($693.8$ G vs. $2081.4$ G).
>
> These results demonstrate that the latency of SS-TPT is a modest and necessary trade-off for achieving strong robustness in safety-critical applications.
>
> * Fig. A: https://anonymous.4open.science/r/qk7/FigA.png
>
> >  **Q4.** Memory-robustness trade-off
>
> To address the reviewer’s concerns, we provide additional analysis on the robustness-memory trade-off averaged over 10 datasets in **Fig. D**.  Prompt-tuning methods (including our SS-TPT) require higher peak memory (2.6 GB) than other lightweight methods (0.9 GB). This overhead is primarily due to the storage of optimizer states and gradient buffers required for the test-time adaptation step.  **As the reviewer rightly pointed out, scaling the number of views ($N$) increases peak memory** because each additional view requires more activations to be retained in memory **to compute gradients during the backpropagation step**. Based on this analysis, we highlight two key observations:
>
> **Higher Robustness at the Same Memory Budget:** While lightweight non-TPT methods are memory-efficient, they fail to provide meaningful adversarial robustness, which limits their practicality in high-security scenarios. In contrast, SS-TPT achieves the most favorable robustness-memory trade-off, consistently delivering higher robustness than TPT-based baselines at identical memory levels.
>
> **Overall Practicality:** When combined with the throughput efficiency shown in **Fig. 1**, SS-TPT offers the most balanced real-world utility. By achieving superior robustness with significantly fewer views (15 views), SS-TPT offers a highly practical defense framework that balances security requirements with resource constraints.
>
> * Fig. D: https://anonymous.4open.science/r/qk7/FigD.png
>
> >  **Q5.** Dependence on prior frameworks
>
> Due to space constraints, please refer to our response to **Reviewer eKBB (Q3)**. Our originality lies in rethinking the recent view in the SOTA methods regarding consistency under adversarial attacks.
>
> >  **Q6.** Discussion of limitations
>
> We thank the reviewer for pointing out this oversight. We will explicitly detail the following technical limitation in the revised manuscript:
>
> Our framework assumes stability and suitability are sufficient indicators of trustworthiness. A potential edge case arises if multiple corrupted augmented views happen to mutually cluster densely and behave consistently. In such rare scenarios, they could artificially yield high SS scores despite being semantically unreliable.
>
> To mitigate these boundary conditions, a natural next step for future work is to broaden the notion of view quality by incorporating complementary metrics, such as spatial cues that capture how consistently local regions behave across views.

---

> > ### Author Rebuttal · Reviewer_VVEX · 2026-04-01
> >
> > I would like to thank the authors for their thorough and high-quality response. In the rebuttal, the authors directly addressed and effectively resolved all of my core concerns. Given that the detailed evidence provided by the authors and their rigorous academic attitude have fully dispelled my previous concerns, I have decided to raise my overall recommendation score from 4 to 5.

---

> > > ### Author Response · Authors · 2026-04-02
> > >
> > > We are deeply grateful to you for reviewing our response and sharing your positive feedback. Your constructive questions regarding latency, memory overhead, and adaptive attack scenarios provided us with an excellent opportunity to evaluate our approach from a different, yet crucial, perspective. We will carefully reflect these detailed analyses and the discussed limitations in our revised manuscript. We sincerely appreciate your decision to raise your score and your support for our paper.

---

### Official Review · Reviewer_SiLA · 2026-03-10

**Soundness:** 3
**Presentation:** 2
**Significance:** 3
**Originality:** 3
**Overall Recommendation:** 4
**Confidence:** 3

**Summary:**

The authors propose SS-TPT (Stability and Suitability-guided Test-time Prompt Tuning) to improve robustness-throughput trade-offs across diverse datasets and varying numbers of views. The idea is to update the text prompt based on two scores: stability - prediction invariance under weak augmentations, and suitability - feature-space density among views.  By reweighting the views,  SS-TPT amplifies the trustworthy views while suppressing corrupted ones. The results show the SOTA performance with a good speed of inference.

**Compliance With Llm Reviewing Policy:**

Affirmed.

**Final Justification:**

I maintain the positive assessment of this work. The rebuttal is thorough and cleared my concerns.

**Key Questions For Authors:**

- Could authors provide more theoretical insights into their proposed SS metrics and update methods?
- Could authors improve their presentation by providing more insights into their method design, since all of them are empirical? e.g., can they consider adding more empirical/theoretical analysis for their proposed method to make it more interpretable?

**Limitations:**

no. See the weaknesses and questions sections.

**Strengths And Weaknesses:**

Strengthens:
1. The methods are well-motivated and easy to follow. The scoring idea is interesting and intuitive. Also, they are supported with comprehensive ablation studies.
2. The results are comprehensive - many results are presented, and convincing, with significant margins of improvement of robustness-efficiency tradeoff.

Weaknesses:
1. The paper has an assumption that views have good and bad qualities, and scoring them (separating good/bad) will help improve the overall robustness. However, this hypothesis may not hold in practice (e.g may be, all views are misleading). How could authors mitigate this case?
2. The scoring system is purely empirical, and I hope authors can provide some theoretical insights into their proposed framework.
3. The presentation could be further improved by incorporating more explanations and details of their techniques. E.g., why use JS divergence for measuring the stability? Why use the power of -1, not other forms of scoring? Why use a convex combination of two normalized scores? How to find the good temperature of SS weights?

---

> ### Author Rebuttal · Authors · 2026-03-30
>
> We thank the reviewer for the insightful questions.
>
> >  **Q1.** Interpretability of design and further analysis + All misleading view case
>
> We first analyze the interpretability of SS-TPT, and then discuss the “all misleading views” case.
>
> **Interpretable Guidance:** An in-depth analysis (**Fig. C**), averaging the SS scores, demonstrates how our scoring mechanism guides attention to the trustworthy views:
>
> 1. **Clean Images:** The unperturbed original view retains the highest SS score, acting as the primary trustworthy view. Notably, actual natural images (ImageNet) receive higher scores (0.52) compared to distribution-shifted variants like sketches in ImageNet-S (0.45) or artistic renditions in ImageNet-R (0.44). This highlights our method's ability to intuitively gauge the naturalness of the source distribution.
>
> 2. **Attacked Images:** Adversarial corruption causes the original view's score to plummet to near zero (e.g., 0.02). SS-TPT then shifts its focus to the augmented views (SS scores > 0.02), which are not directly perturbed, identifying them as the new trustworthy views.
>
> **The "All Misleading" Worst-Case:** If *all* views are misleading, standard adaptation fails. However, unlike prior methods that treat all views equally, SS-TPT mitigates this failure mode by identifying the *relatively* trustworthy view. We empirically simulate this worst-case using an extreme scarcity setting with only 1 augmented view (**Tab. C**), where visual diversity is minimal and the chance of all views (original view+one augmented view) being misleading is highest. Even under this extreme setting, SS-TPT yields a **+12.1%p** Rob. improvement over baselines.
>
> Thus, our SS scores not only make the adaptation interpretable but also provide a critical advantage even when all views can be misleading. We will include this discussion in the revised manuscript.
>
> * Fig. C: https://anonymous.4open.science/r/qk7/FigC.png
>
> * Tab. C: https://anonymous.4open.science/r/qk7/TabC.png
>
> >  **Q2.** Lack of theoretical insight + Grounding of SS metrics
>
> We agree that providing theoretical insights can further improve the manuscript. Our method is guided by a simple principle: test-time adaptation should rely more on views that are locally stable and well-supported by other views, and less on isolated or unstable views. Based on this principle, we provide a theoretical interpretation of our SS-guided consistency loss.
>
> Concretely, the stability score (Eq. 4) acts as a zero-order measure of local predictive invariance under weak perturbations, while the suitability score (Eq. 5) measures how centrally a view lies within the multi-view feature set. The SS-guided consistency loss
> $$
> L_{scons} = \frac{1}{|B|} \sum_{x_i \in B} D_{KL}(p(\cdot \mid x_i) \| \hat{p}^{(-i)})
> $$
> can then be interpreted as a reliability-weighted consensus regularizer, where $ \hat{p}^{(-i)}$ is calculated using the stability and suitability scores (Eq. 8). Accordingly, $L_{scons}$ downweights isolated or unstable views and emphasizes views that are both locally consistent and supported by neighboring views.
> Compared with TPT and R-TPT, which apply an unweighted consistency term and no consistency term, respectively, our SS-TPT applies consistency selectively based on view reliability. From a theoretical perspective, our method lies between the two extremes of uniform consistency in TPT and no consistency in R-TPT. We will clarify this theoretical positioning in the revised manuscript.
>
> >  **Q3.** Explanation of design choices
>
> **JS Divergence:** Stability compares an original prediction and its augmented variant. Since neither is an absolute "ground truth," the distance metric should be symmetric (e.g., JS or symmetric KL). We use JS in Eq. 4 because it is stable (bounded $[0, 1]$) and performs slightly better than symmetric KL (**Tab. D(b)**). Conversely, adaptation (Eq. 8) requires directional KL because the goal is to pull the prediction toward a trustworthy view.
>
> **Power of -1:** It can robustly convert a distance into a quality score. Unlike exponential decay ($e^{-d}$) which vanishes too rapidly, $(d+\eta)^{-1}$ provides a heavy-tailed distribution. This prevents the scores of moderately corrupted views from collapsing to zero prematurely, allowing smoother weighting.
>
> **Convex Combination:**  Before combination, Min-Max strictly bounds both scores to $[0, 1]$, guaranteeing a comparable scale. Z-score can yield unbounded negative values, which hurts performance (**Tab. D(c)**). The convex combination elegantly balances two properties: local smoothness (stability) and global manifold density (suitability). As shown in **Tab. 6**, they are highly complementary.
>
> **Temperature Tuning:** **Fig. B** shows robustness peaks at $\tau_w=0.25$ but remains stable over a broad range. This indicates that SS-TPT is robust to the choice of temperature.
>
> * Tab. D: https://anonymous.4open.science/r/qk7/TabD.png
>
> * Fig. B: https://anonymous.4open.science/r/qk7/FigB.png

---

> > ### Author Rebuttal · Reviewer_SiLA · 2026-04-01
> >
> > Thanks for the rebuttal - it is very detailed. My concerns are resolved. I do not have other questions.

---

> > > ### Author Response · Authors · 2026-04-02
> > >
> > > Thank you for acknowledging our rebuttal and for your valuable time. We are glad that our detailed analysis successfully resolved your questions. Your constructive feedback has allowed us to further strengthen our manuscript, specifically by highlighting the theoretical insights and design choices of SS-TPT more clearly. We will certainly incorporate all of these details into the final revision. We sincerely appreciate your support.

---

### Official Review · Reviewer_eKBB · 2026-03-12

**Soundness:** 3
**Presentation:** 3
**Significance:** 3
**Originality:** 3
**Overall Recommendation:** 4
**Confidence:** 3

**Summary:**

The paper tackles the challenge of defending VLMs against adversarial perturbations during inference. Existing defensive Test-Time Adaptation (TTA) methods suffer from a severe computational bottleneck and a robustness-throughput trade-off. The authors propose SS-TPT to tackle this issue by evaluating the quality of each augmented view rather than treating them equally. View quality is measured using two scores: Stability and suitability. Extensive experimental results show superior robustness-throughput trade-offs across multiple datasets.

**Compliance With Llm Reviewing Policy:**

Affirmed.

**Final Justification:**

Based on the overall contributions of the paper and the feedback from other reviewers, I maintain my recommendation of “Weak Accept.”

**Key Questions For Authors:**

1. How sensitive is the stability score to the exact choice of weak augmentations? Could a deliberately designed adaptive attack hijack the stability score?
2. How many weak stochastic augmentations are required to calculate the stability score per view?

**Limitations:**

The authors have not adequately discussed their technical limitations. While they provided an "Impact Statement," they defer limitations by simply stating: "We encourage its use alongside careful evaluation, transparent reporting of limitations, and complementary safety practices to ensure responsible deployment" without actually reporting those explicit technical limitations, such as edge cases where suitability clustering fails, within the main text.

The authors included a dedicated Impact Statement highlighting positive societal impacts, notably that reducing the number of views lowers test-time computation costs, thereby contributing to lower energy consumption and more environmentally sustainable VLM deployments.

**Strengths And Weaknesses:**

The main strengths include: (1) The manuscript is well-written and motivated. (2) The dual-metric approach is mathematically and intuitively sound. Combining stability and suitability checks is a robust heuristic to isolate clean semantic features from adversarial noise. (3) The approach shows good practical significance by consistently delivering superior robustness-throughput trade-offs across a wide range of augmented view counts.

The weaknesses include: (i) Calculating the stability score implies hidden forward-pass overhead, and the extra compute needed to calculate stability per view could offset these gains in extreme edge deployments. (2) A more explicit breakdown of the FLOPs/latency per module would strengthen the presentation. (3) The adaptation is strictly limited to an instance-level setup and its significance in a continuous streaming video setting is less clear. (4) Consistency losses and entropy minimization are standard in TTA, and the proposed architectural model is heavily based on intelligent weighting of these standard tools.

---

> ### Author Rebuttal · Authors · 2026-03-30
>
> We thank the reviewer for the constructive feedback.
>
> >  **Q1.** Cost of stability scoring + Lack of detailed latency breakdown
>
> Our Stability and Suitability (SS) strategy is highly efficient due to:
>
> **(1) Selective Computation:** The stability calculation does not scale linearly with the total number of views. Following the protocol of TPT and R-TPT, we first filter for a low-entropy subset $\mathcal{B}$. The stability score is evaluated only on these confident views, which strictly limits the overhead. As shown in **Fig. A**, stability scoring takes a fraction of the pure processing time (0.0383 sec).
>
> **(2) Superior Performance with Fewer Views:** Guided by SS scores, SS-TPT achieves superior robustness (81.0%) with only 16 views, whereas the previous state-of-the-art R-TPT requires 64 views to reach just 79.8%. Total inference time is heavily dominated by data augmentation. By drastically reducing the required views, SS-TPT significantly cuts down both the total latency (1.67 sec $\rightarrow$ 0.74 sec) and the computational cost (2081.4 G $\rightarrow$ 693.8 G FLOPs), as shown in **Fig. A**.
>
> This confirms that investing a fraction of compute for high-quality views is a highly effective trade-off, **resulting in a system that is simultaneously faster, lighter, and more robust.**
>
> * Fig. A: https://anonymous.4open.science/r/qk7/FigA.png
>
> >  **Q2.** Instance-level limitation and video-streaming significance
>
> While focused on instance-level adaptation for VLMs, SS-TPT naturally has two key properties for continuous video streaming:
>
> **High Throughput Potential:** Real-time video requires low per-frame latency. Unlike prior methods with massive augmentation (e.g., 64 to 256 views), Fig. 1 and Fig. A show that SS-TPT achieves superior robustness with only 16 views, substantially reducing latency and FLOPs per frame, which makes it suitable for real-time streaming.
>
> **Extension to Temporal Views:** In a video stream, adjacent frames naturally act as temporal "augmented views." Extending our SS scoring to evaluate these temporal frames, instead of generating augmentations, would essentially eliminate the augmentation overhead entirely.
>
> Thus, SS-TPT's efficient design and unique view quality assessment mechanism naturally extend to streaming scenarios, making this a highly promising direction for future work.
>
> >  **Q3.** Dependence on standard TTA
>
> While built upon TTA components such as consistency losses and entropy minimization, our originality lies in rethinking the recent view in the SOTA methods regarding consistency under adversarial attacks. This can be summarized into three paradigms:
>
> **Unguided Consistency (TPT)**: The foundational method, TPT, minimizes marginal entropy **(Eq. 2)**, which implicitly includes an unguided KL consistency term that enforces alignment across all views, regardless of their individual quality.
>
> **No Consistency (R-TPT)**: R-TPT argued that enforcing consistency with corrupted adversarial views actively misleads adaptation. Consequently, R-TPT completely removed the KL term, relying exclusively on point-wise entropy **(Eq. 3)**.
>
> **SS-Guided Consistency (Ours)**: Our originality stems from discovering that consistency itself is not the problem. We challenge the R-TPT paradigm by reinstating the KL term. Using our SS scores, we formulate an SS-guided consistency loss **(Eq. 8)** and a combined adaptation objective. This ensures the model aligns only with trustworthy references, avoiding corrupted views.
>
> By guiding consistency rather than blindly discarding it, SS-TPT achieves superior adversarial robustness and preserves high clean accuracy, all while requiring significantly fewer views than prior methods.
>
> >  **Q4.** Number of augmentations + Sensitivity + Adaptive attack for stability
>
> **Minimal Augmentations:** To compute stability for each view ($x_i$), we require only one stochastically augmented variant ($\tilde{x}_i$), which keeps the computational overhead minimal. Each variant is generated using a composition of four augmentations. (Details in Sec. C.)
>
> **Low Sensitivity to Augmentation Choice:** The stability score is highly robust to the exact hyperparameters of the weak augmentations. **Tab. A** demonstrates that scaling augmentation intensity or removing individual augmentation components causes only negligible performance drops.
>
> **Adaptive Attacks:**  Due to space constraints, please refer to our response to **Reviewer vveA (Q4)** and **Tab. B**, which demonstrates that SS-TPT maintains superior resilience even under strong adaptive attacks.
>
> * Tab. A: https://anonymous.4open.science/r/qk7/TabA.png
>
> >  **Q5.** Discussion of limitations
>
> Due to space constraints, please refer to our response to **Reviewer VVEX (Q6)**. We discuss the rare edge case where multiple corrupted views might consistently cluster and propose incorporating spatial cues in future work to further refine trustworthiness.

---

> > ### Author Rebuttal · Reviewer_eKBB · 2026-04-03
> >
> > I thank the authors for their detailed response. My questions have been adequately addressed, and I have no further comments. Based on the overall contributions of the paper and the feedback from other reviewers, I maintain my recommendation of “Weak Accept.”

---

> > > ### Author Response · Authors · 2026-04-03
> > >
> > > Thank you for your thoughtful acknowledgement of our rebuttal. We are pleased that our clarifications on the SS-guided consistency loss and efficiency trade-offs effectively resolved your questions. Your feedback was instrumental in helping us better articulate the distinction between our approach and standard TTA methods, as well as providing a clearer roadmap for future video-streaming applications. We are committed to incorporating these comprehensive analyses into the final revised manuscript. We truly appreciate your support.

---

### Decision · Program_Chairs · 2026-04-30

**Decision:**

Accept (regular)

**Comment:**

In this submission, the authors proposed SS-TPT which addresses the robustness-throughput trade-off in test-time adaptation for VLMs by scoring augmented views on stability and suitability, enabling strong adversarial robustness with far fewer views than prior methods. All four reviewers found the work technically sound, well-motivated, and experimentally thorough. Post-rebuttal, all concerns, including computational cost, generalization to other VLMs, adaptive attacks, and limitations discussion,  were fully resolved, with one reviewer upgrading their score. The AC also recommends acceptance.